# An extensive disulfide bond network prevents tail contraction in *Agrobacterium tumefaciens* phage Milano

Ravi R. Sonani [1], Lee K. Palmer[2], Nathaniel C. Esteves[3], Abigail A. Horton[3], Amanda L. Sebastian[3], Rebecca J. Kelly[3], Fengbin Wang [1,5], Mark A. B. Kreutzberger[1], William K. Russell [2], Petr G. Leiman [4] ✉, Birgit E. Scharf [3] ✉ & Edward H. Egelman [1] ✉

A contractile sheath and rigid tube assembly is a widespread apparatus used by bacteriophages, tailocins, and the bacterial type VI secretion system to penetrate cell membranes. In this mechanism, contraction of an external sheath powers the motion of an inner tube through the membrane. The structure, energetics, and mechanism of the machinery imply rigidity and straightness. The contractile tail of *Agrobacterium tumefaciens* bacteriophage Milano is flexible and bent to varying degrees, which sets it apart from other contractile tail-like systems. Here, we report structures of the Milano tail including the sheath-tube complex, baseplate, and putative receptor-binding proteins. The flexible-to-rigid transformation of the Milano tail upon contraction can be explained by unique electrostatic properties of the tail tube and sheath. All components of the Milano tail, including sheath subunits, are crosslinked by disulfides, some of which must be reduced for contraction to occur. The putative receptor-binding complex of Milano contains a tailspike, a tail fiber, and at least two small proteins that form a garland around the distal ends of the tailspikes and tail fibers. Despite being flagellotropic, Milano lacks thread-like tail filaments that can wrap around the flagellum, and is thus likely to employ a different binding mechanism.

Milano is a contractile tail bacteriophage that infects the pathogenic plant bacterium *Agrobacterium tumefaciens*, an important tool that is widely used in genetic manipulations of plants[1]. Milano is similar to *Agrobacterium* sp. H13-3 phage 7-7-1 as their proteins share ~89% sequence identity, and are closely related morphologically[2,3]. Both, Milano and 7-7-1 require the bacterial host to carry actively rotating flagella to which they bind to initiate the infection process[2] (Fig. 1a). The phage particle is likely to experience a significant mechanical stress during this process[4–6]. Generally, flagellotropic phages possess long and curly tail or head fibers, which can wrap around the flagellar filament and allow the phage to travel along the flagellum toward the host cell surface as the flagellum rotates[3]. However, no such fibers have been identified in Milano (Fig. 1b).

The Milano genome contains genes commonly found in other contractile tail-like injection systems (CIS). Gene *20* encodes a putative contractile sheath (gene product, gp20), gene *21*—a putative inner tube

[1]Department of Biochemistry and Molecular Genetics, University of Virginia School of Medicine, Charlottesville, VA 22903, USA. [2]Mass Spectrometry Facility, University of Texas Medical Branch, Galveston, TX 77555, USA. [3]Department of Biological Sciences, Virginia Tech, Blacksburg, VA 24061, USA. [4]Department of Biochemistry and Molecular Biology, University of Texas Medical Branch, Galveston, TX 77555, USA. [5]Present address: Department of Biochemistry and Molecular Genetics, University of Alabama at Birmingham, Birmingham, AL 35233, USA. ✉e-mail: pgleiman@utmb.edu; bscharf@vt.edu; egelman@virginia.edu

**Fig. 1 | Flagellotropic phage Milano possesses a flexible tail. a** Spot assay of Milano on *Agrobacterium tumefaciens* C58 wild type, BM140 (DflaA–D)−flagella-minus (fla⁻) strain[65], and PMM5 (DmotA)−the non-motile (mot⁻) strain[65]. Numbers (n) indicate the Milano phage dilution ($10^n$) that was spotted. The Milano could infect and lyse the wild type *A. tumefaciens* C58 cells up to the stock dilution of $10^{-8}$, while it could not infect and lyse the fla⁻ and the mot⁻ cells suggesting its dependency on the rotating flagellar filament. Very weak clearance of the fla⁻-bacterial lawn at high phage concentrations is potentially due to the fact that phage particles are able to adhere to the flagellar hook. **b** Cryo-EM image of Milano bacteriophage and 2D class averages of the bacteriophage tail. The black and white arrowheads indicate the curvature in the tail. The scale bar is ~50 nm. Total of 7900 cryo-EM micrographs were collected showing the varying degrees of curvature in Miano's tail. **c** Cryo-EM density map of Milano curved tail with tube and sheath region colored in orange and green, respectively. **d** Cryo-EM density map of Milano baseplate-proximal tail segment having C6 symmetry coaxial with baseplate axis, colored similar to (**c**). **e** The 3D ribbon model of Milano tail tube protein, gp21, and its organization into the helical tail-tube. The blue loop (residues 44−61) of gp21 invades the neighboring subunit during tube oligomerization. Cyan ball indicates the site (residues 111−112), where a loop present in other CIS is missing in Milano gp21. Cys residues involved in intra-hexamer and inter-hexamer disulfide bonds are shown in yellow and green, respectively. Grooves on the tube surface are shown by red arrows. **f** The 3D ribbon model of Milano sheath protein, gp20. The sheath handshake (SHD), sheath-body (SBD), and sheath extension (SED) domains are colored orange, white, and green, respectively. Cys involved in inter-subunit disulfide bonds are highlighted in red. **g** The oligomeric arrangement of gp20 in the Milano sheath. Two *n* subunits donate their N- and C-terminal strands (part of SHD) to the SHD of *n*−1 subunit, known as the β-sheet augmentation mechanism and reminiscent of "hand shaking", stabilizing the sheath assembly.

(gp21) of tail, and genes *26* and *27* encode the hub and baseplate central spike, respectively[7]. However, the typical genomic organization of the Milano tail is not translated into a standard architecture. As dictated by their six-fold symmetry and energetics, virtually all previously studied CISs are straight on a 100-nm scale[8–22]. The tails of Milano and *Listeria* phage A511[23] form a rare exception because they can bend to varying degrees when imaged in neutral buffer conditions (Fig. 1b). How such bent and flexible tails can pierce the cell envelope is unknown.

To reveal the unique host recognition apparatus of Milano and to understand the structural origin of its tail flexibility, we have determined the atomic structures of Milano tail, baseplate, and receptor-binding complex by cryo-electron microscopy (cryo-EM). We show that protein subunits forming the Milano tail sheath and baseplate are covalently linked by multiple disulfide bonds into a shear-proof assembly, which might allow the phage to survive the mechanical stress caused by binding to a flagellum rotating through the viscous media surrounding the host. The structure of the Milano receptor binding complex is unique in its organization and the manner of attachment to the baseplate. It is also devoid of any filaments that could wrap around a flagellum. Additionally, we

determined the structure of the tail tube prior to and after tail sheath contraction and found that it is virtually unchanged, providing the first experimental proof for a long-standing assumption that the tail tube has the same conformation in the pre-attachment (extended sheath) and post-attachment (contracted sheath) states.

## Results

### Overview of the cryo-EM image reconstruction procedure

The three major components of the Milano tail—the inner tube, the outer sheath, and the baseplate with receptor-binding proteins (RBPs)—are multiprotein assemblies, which display different symmetries. We used a multi-faceted cryo-EM approach to obtain the structure of different regions of the tail by applying the highest allowed symmetry.

In the initial extended (pre-attachment) conformation, virtually all tails in cryo-EM images were bent (Fig. 1b), complicating the reconstruction procedure. In the contracted (post-attachment) state, the tails were straight (Fig. 2a; Supplementary Fig. 1a, b). The curved tail structure was reconstructed by dividing the tail into overlapping segments. Fragments with a common curvature were clustered using 3D

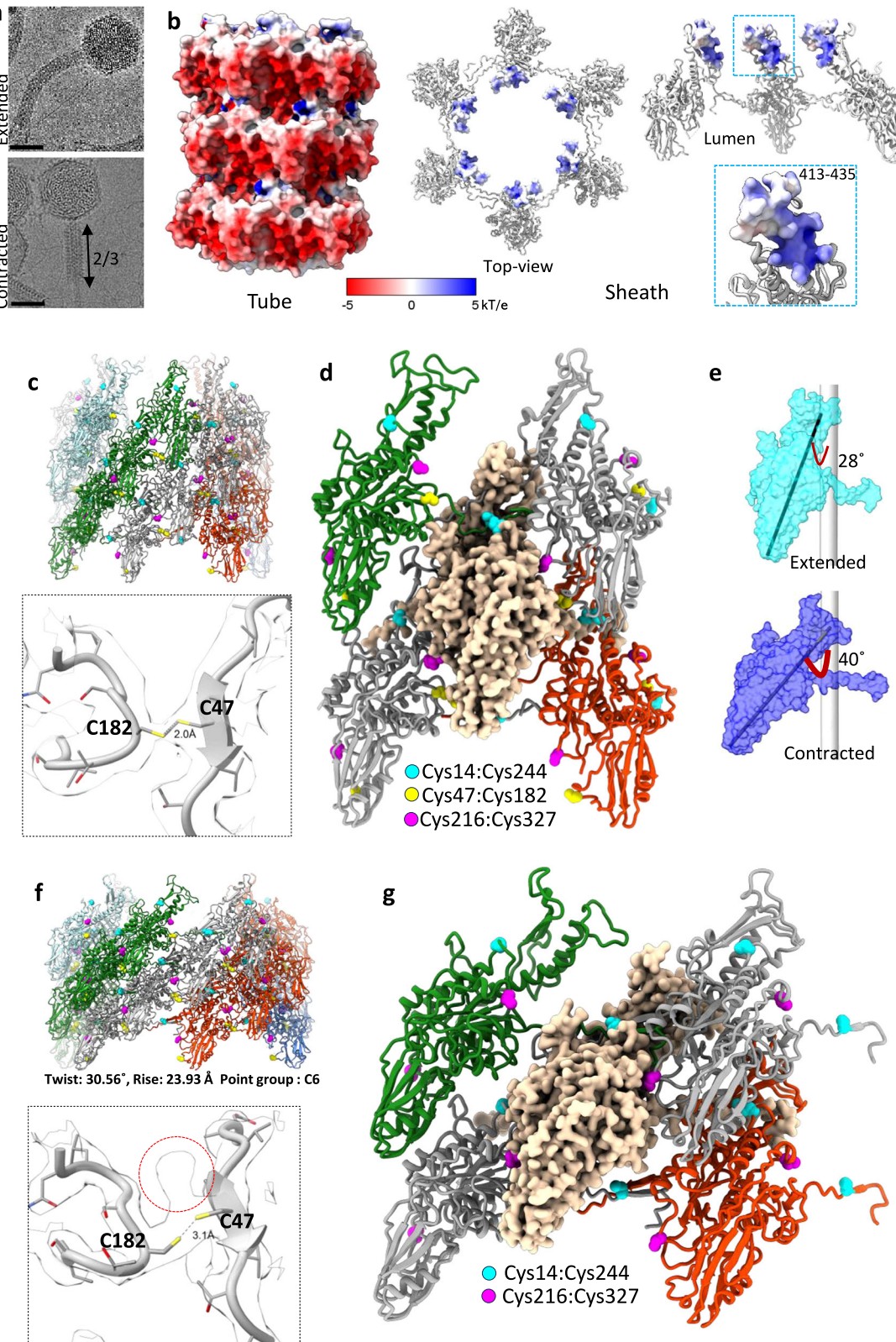

**d**
○ Cys14:Cys244
● Cys47:Cys182
● Cys216:Cys327

**f**
Twist: 30.56°, Rise: 23.93 Å  Point group : C6

**g**
○ Cys14:Cys244
● Cys216:Cys327

variability analysis and used for reconstruction with no symmetry imposed (Fig. 1c). A similar procedure was used previously in the reconstruction of bacterial flagellar filaments[24]. The baseplate-proximal segments of the sheath and tube were coaxial with the baseplate. They were reconstructed with C3 symmetry and then used as a straight tail for comparison with a curved tail segment (Fig. 1d; Supplementary Fig. 2a).

## Atomic structure of the Milano sheath and tube in the extended state

Milano tail tube protein, gp21 adopts a β-sandwich structural fold similar to other known CIS tube proteins (Fig. 1e; Supplementary Fig. 2c)[9,10,16,21,25-30]. It oligomerize into a continuous 24-stranded β-barrel hexamer, where a loop (residues 44–61) interacts with the neighboring subunit. The hexamers stack on top of each other with a twist of

**Fig. 2 | Contraction of Milano tail. a** Cryo-EM image of Milano tail in an extended and contracted state. The tail becomes straight, and the sheath is reduced to 2/3 of the extended length. The scale bar is ~50 nm. Total of 7900 and 10,800 cryo-EM micrographs of Milano were collected in extended and contracted states, respectively. **b** The electrostatic potential of the Milano tube outer surface, and sheath-region (residues 413–435) interacting with the tube. **c** Milano tail sheath in pre-contraction (extended) state. Protomers are colored by right-handed protofilaments. **d** A sheath subunit interacting with four neighbors, two in each right- and left-handed protofilaments. Each subunit of Milano forms three different disulfide bonds with its neighboring subunits, for a total of six disulfide bonds per subunit.

Close-up view of C47:C182 disulfide bond between subunits of right-handed protofilament is shown as an inset. **e** Comparison of tilting of sheath subunit with respect to the central axis in extended and contracted states. The central axis is shown by gray cylinders. **f** Tail sheath in contracted state colored similar to (**c**). The contracted sheath adopts a different helical symmetry (twist: 30.56°, rise: 23.93 Å, C6 point group). **g** A sheath subunit interacting with four neighbors in a contracted state. Contracted sheath preserves two out of three inter-subunit disulfide bonds. The broken disulfide bond, Cys47:Cys182 after contraction is shown with a density map as an inset. The density blob shown by the red circle possibly accounts for the DTT molecule bound to the Cys side chain.

28.21° and a rise of 34.07 Å resulting in a hollow helical tube with a ~40 Å diameter lumen (Fig. 1e; Supplementary Fig. 2b). The inter-ring contact is mediated by the N-terminal end (residues 1–7) and 44–61 loop as removal of which would result in no contact between rings (Supplementary Fig. 2b). Each gp21 subunit forms a disulfide bond (Cys43–Cys82) with a neighbor within the same hexamer and a disulfide bond (Cys3–Cys19) with a subunit of the adjacent hexamer (Fig. 1e; Table 1; Supplementary Fig. 3), resulting in a fully disulfide-

crosslinked tubular structure. The inter-ring disulfide bond (Cys3–Cys19) in Milano tube seems to mimic the C-terminal extension invading the upper hexamer in the tubes of Siphophage, 80α and SPP1 that strengthen the inter-ring association. The presence of this disulfide bond provides increased stability to the Milano tube as it increases the free energy of a two-ring complex from 32.4 to 52.1 kCal/mol as calculated by PISA[31].

The Milano sheath wraps tightly around the tube with both assemblies having the same helical symmetry in the extended state (Supplementary Fig. 2a). The Milano sheath subunit gp20 adopts the consensus CIS sheath protein structure as it consists of three domains —an inner Sheath Handshaker Domain (SHD, residues 1–23 and 381–503), a middle globular sheath body domain (SBD, residues 24–90 and 204–380), and an external sheath extension domain (SED, residues 91–203) (Fig. 1f; Supplementary Fig. 2d). Each sheath subunit donates two long linker arms at their N- and C-termini (residues 1–23 and 486–503, respectively) to two other subunits, where they become integral components of those subunits' SHDs by a β-sheet augmentation mechanism[32] (Fig. 1g). As a result, all sheath subunits are linked into a sleeve-like mesh.

Both the structure of the subunit and the topology of the mesh in the Milano sheath are similar to those found in other CIS[19,25,26,28,29,33]. The importance of the mesh's integrity and native topology for sheath contraction has been studied in detail for the T6SS and R-type pyocin[8,34]. The high degree of structural conservation suggests that the Milano sheath and all other CIS sheaths are likely to have similar properties. T4 bacteriophage possesses a domain decorating the sheath beyond the SED[19] (Supplementary Fig. 2d). These domains likely modulate the degree of sheath compaction or shortening upon contraction.

### Structural attributes responsible for tail flexibility

Compared to other CIS tube proteins[25–29], the Milano tube protein lacks a loop between residues 111 and 112 that would otherwise be present and anchoring the hexameric rings tightly (Fig. 1e; Supplementary Fig. 2c). As a result, the surface of the tube displays substantial grooves at the interface of two rings that are not found in straight tubes of other CIS (Fig. 1e; Supplementary Fig. 4). In the curved tail, these grooves are smaller on the inner surface of the curve when compared to the straight state, showing that these grooves enable the bending (Supplementary Fig. 2e). Due to the absence of said loop in Milano, the interfacial area (~8500 Å²) between two hexamer rings is reduced to ~55% of that of other CIS tubes i.e. T4 (~16,600 Å²), T6SS (~14,600 Å²), RTP (~15,100 Å²), PVC (~16,100 Å²), and AFP (~15,900 Å²) as calculated by PISA[31]. Flexible tails of Siphophages (non-contractile tail phages), such as SPP1, 80α and YSD1[35–38], which are homologous to inner tubes of contractile tail phages[39], display similar grooves showing the correlation of the presence of grooves with flexibility (Supplementary Fig. 4). However, the loop, which is absent in the Milano tube (between residues 111 and 112), is present in these phages (Supplementary Fig. 2c), suggesting that the structural origin of grooves in these bacteriophages is different from that of Milano.

The backbone traces of the tube subunit in the baseplate-proximal straight region and in the curved middle region of the

## Table 1 | Inter-subunit disulfide bonds within and between structural parts of Milano

| *Within sub-structure* | | |
|---|---|---|
| Tube | gp21, Cys3 | gp21, Cys19 |
| | gp21, Cys43 | gp21, Cys82 |
| Sheath | gp20, Cys14 | gp20, Cys244 |
| | gp20, Cys47 | gp20, Cys182 |
| | gp20, Cys216 | gp20, Cys327 |
| Base-plate wedge | BW1 (gp28), Cys3 | BW3 (gp30), Cys22 |
| | BW1 (gp28), Cys22 | BW3 (gp30), Cys75 |
| | BW2_outer (gp29), Cys68 | BW3 (gp30), Cys23 |
| | BW2_outer (gp29), Cys282 | BW3 (gp30), Cys206 |
| | BW2_outer (gp29), Cys302 | BW3 (gp30), Cys130 |
| | BW2_outer (gp29), Cys395 | BW2_Inner (gp29), Cys234 |
| | BW2_outer (gp29), Cys3 | BW2_inner (gp29), Cys68 |
| | BW2_outer (gp29), Cys234 | BW2_inner (gp29), Cys282 |
| Tail spike | TSP (gp124), Cys24 | TSP (gp124), Cys24 |
| | TSP (gp124), Cys24 | TSP (gp124), Cys26 |
| | TSP (gp124), Cys26 | TSP (gp124), Cys26 |
| Short tail fiber | STF (gp31), Cys29 | STF (gp31), Cys67 |
| | STF (gp31), Cys84 | STF (gp31), Cys102 |
| | STF (gp31), Cys127 | STF (gp31), Cys144 |
| *Between sub-structures* | | |
| Tube—baseplate centerpiece | Tube (gp21), Cys19 | BCP (gp25), Cys5 |
| Sheath—baseplate wedge | BW1 (gp28), Cys32 | Sheath (gp20), Cys14 |
| | BW1 (gp28), Cys74 | Sheath (gp20), Cys501 |
| | BW1 (gp28), Cys29 | Sheath (gp20), Cys501 |
| Sheath—long tail spike | Sheath (gp20), Cys327 | TSP (gp124), Cys15 |
| Baseplate wedge—tail spike | BW1 (gp28), Cys41 | TSP (gp124), Cys15 |
| | BW3 (gp30), Cys69 | TSP (gp124), Cys15 |
| Baseplate wedge—short-tail fiber | BW2_Inner (gp29), Cys302 | STF (gp31), Cys38 |
| | BW2_Inner (gp29), Cys224 | STF (gp31), Cys38 |
| | BW2_Inner (gp29), Cys367 | STF (gp31), Cys38 |
| | BW2_Outer (gp29), Cys367 | STF (gp31), Cys38 |
| | BW2_Outer (gp29), Cys224 | STF (gp31), Cys38 |
| | BW3 (gp30), Cys147 | STF (gp31), Cys38 |
| Baseplate centerpiece—baseplate wedge | BCP (gp25), Cys318 | BW2_Inner (gp29), Cys3 |

Milano tail structure are virtually identical with an RMSD value between their Cα atoms of ~0.5 Å which is at the level of experimental error. The curvature is therefore realized at the level of side chains, which adopt slightly different conformations in the grooves separating the subunits. The interactions of sheath and tube subunits in straight and curved tail segments are very similar, suggesting that these interactions do not constrain the bending (Supplementary Fig. 2e).

### Structure of the tube in the contracted tail

The structure of the tube in the contracted tail has long been assumed to be similar or identical to that in the extended tail[7,8]. However, this has never been established experimentally. The flexibility of the Milano tube in the extended tail made the question of the tube structure in the contracted state even more intriguing.

We first attempted to reconstruct the structure of the tube in the contracted tail by boxing out the extruded part of the tube (Fig. 2a) but found too few such particles to generate a high-resolution reconstruction. We therefore sought to use the tube within regions that are still surrounded by the contracted sheath. However, the tube and the contracted sheath have different helical symmetries. After contraction, when the sheath shortens to ~70% of its extended length, the ratio of sheath:tube mass in projected images increases to more than 4:1. This hinders unambiguous indexing of tube symmetry from power spectra. We improved this ratio to ~2:1 by windowing the projected images to only include the central part of the image comprising the tube and the portion of the outer sheath superimposed onto it. The resultant power spectrum showed the additional layer lines from the tube, which allowed us to confirm that the tube symmetry had not changed from the extended state (Supplementary Fig. 5). During 3D reconstruction, we removed the sheath contribution by an iterative, three-dimensional masking procedure while applying the helical symmetry of the tube. This approach led to a tail tube reconstruction in the contracted state.

The structures of individual tail tube subunits in the extended and contracted states could be superimposed with an RMSD of ~0.7 Å between 127 pruned Cα atoms. This is probably at the level of experimental error in the determination of Cα positions at this resolution. If we compare a stack of three hexamers in the tube rather than individual subunits, there is an RMSD of ~1.6 Å (Supplementary Fig. 1c, d). This analysis shows that the overall structure of individual subunits and the tube as a whole are indeed the same in the extended and contracted states, as had previously been assumed.

Notably, the tube straightens after contraction, suggesting that it becomes rigid and suitable for cell-membrane piercing (Fig. 2a; Supplementary Fig. 1a, b). This raises the question of what causes the flexible-to-rigid transformation upon contraction in the absence of any apparent conformational changes within the tube subunit or the hexamer. Our results suggest that the answer lies in the distribution of electrostatic potential on the tube's outer surface.

The Milano tube carries a substantial negative charge on its outer surface resulting in bands of negative electrostatic potential (Fig. 2b). In the extended tail, these charges are neutralized by the sheath (specifically, by the SHD), and the tube-sheath assembly is flexible (Fig. 2b). In the contracted tail, the tube no longer interacts with the sheath, and the tube's surface charges repel each other and straighten the tube. A similar phenomenon has been demonstrated in polymer-electrolyte physics—a polymer with charged functional groups on its surface adopts a straight conformation because of same-sign charge repulsion[40,41]. In other CIS, tight packing of tube subunits may play a more important role in tube rigidity than the surface electrostatics (e.g. in RTP and PVC), but the contribution of the surface charges might nevertheless be critical for some tubes (e.g. some T6SSs) (Supplementary Fig. 4).

### Sheath protomers are locked by covalent disulfide-bonds

In contrast to all other CIS previously studied at high resolution, Milano sheath subunits are inter-locked covalently by disulfide bonds. The cryo-EM map shows that each sheath-subunit forms three different disulfide bonds with neighboring subunits, covalently linking all subunits together (Fig. 2c, d; Table 1; Supplementary Fig. 3). One neighboring subunit along a left-handed 6-start helix is linked by one disulfide bond (Cys14$_{SHD}$:Cys244$_{SBD}$), while the neighboring subunit along the right-handed 6-start helix is linked by two disulfide bonds (Cys47$_{SBD}$:Cys182$_{SED}$ and Cys216$_{SBD}$:Cys327$_{SBD}$) (Fig. 2c, d), with the subscripts referring to the sheath protein domains. To confirm these disulfide bonds, we have conducted extensive disulfide mapping using mass spectrometry, where we confirmed the Cys14$_{SHD}$:Cys244$_{SBD}$ and Cys47$_{SBD}$:Cys182$_{SED}$ linkages, with both showing excellent sequence coverage (Supplementary Fig. 6; Supplementary Table 1). Mass spectrometry identified an additional disulfide bond Cys467$_{SHD}$:Cys501$_{SHD}$ which is not clearly apparent in our cryo-EM map.

Such a net of disulfide bonds makes the Milano tail resistant to contraction by chemical inducers such as urea or guanidine hydrochloride that cause contraction in other systems[23,42–44], and we failed to induce any contractions using these agents alone. We could induce contraction by a mixture of guanidine hydrochloride (3M) and dithiothreitol (DTT, 5 mM), a reducing agent. Of note, the Milano baseplate also contains intersubunit disulfide bonds that must be reduced to allow sheath contraction (see below).

To explore the fate of sheath disulfide bonds upon contraction, we solved the atomic structure of the contracted sheath (Fig. 2f, g). The structures of the sheath subunit in the contracted and extended states are similar and can be superimposed with an RMSD of 0.7 Å between 347 pruned Cα atoms out of a total of 491 Cα atoms with an RMSD of 3.9 Å. The larger all-atom RMSD arises from the different relative orientations of the linker arms invading the neighboring subunits' SHD. The rearrangement of sheath subunits upon contraction resembles that found in other CISs (roughly, each subunit rotates and moves radially), and the integrity of the two-dimensional mesh created by the sheath linker arms is preserved. The conservation of these properties suggests that the Milano sheath contracts in a domino wave-like manner similar to the pyocin sheath[8]. However, upon contraction the length of the Milano sheath decreases by only 30% compared to 60–75% in other CIS[23,25,33,42] (Fig. 2a). Accordingly, the motion of subunits in the Milano sheath during contraction is much reduced compared to that in other CISs. Upon full contraction, the Milano sheath subunit rotates by ~12° and moves by ~13 Å radially (Fig. 2e; Supplementary Videos 1 and 2). In contrast, the radial movements in pyocin[16] and T4[33] are ~35 and ~60 Å, respectively.

Two of the three disulfide bonds that crosslink sheath subunits to each other are retained in the contracted sheath (Cys14$_{SHD}$–Cys244$_{SBD}$ and Cys216$_{SBD}$–Cys327$_{SBD}$) whereas the Cys47$_{SBD}$–Cys182$_{SED}$ bond is broken (Fig. 2c, d, f, g). This suggests that the disulfide bond Cys47$_{SBD}$–Cys182$_{SED}$ needs to be released to allow for the rotation and radial movement of sheath subunits during contraction. These cysteine residues are found conserved in *Agrobacterium* infecting bacteriophage clade, which includes bacteriophage 7-7-1[45] (Uniprot id: J7FA80, https://www.uniprot.org/uniprotkb/J7FA80/entry) and OLIVR4 (Uniprot id: A0A858MST0, https://www.uniprot.org/uniprotkb/A0A858MST0/entry) (Supplementary Fig. 7). This indicates the functional importance of these disulfides in *Agraobacterium* phages.

### Structure and disulfide bond network of the Milano baseplate

We have determined the structure of the distal part of the tail comprising the baseplate with associated receptor-binding proteins (RBPs) and two baseplate-proximal layers of the sheath-tube complex to ~3.2 Å resolution (Fig. 3). To describe the locations and putative functions of Milano baseplate proteins, we will use a nomenclature

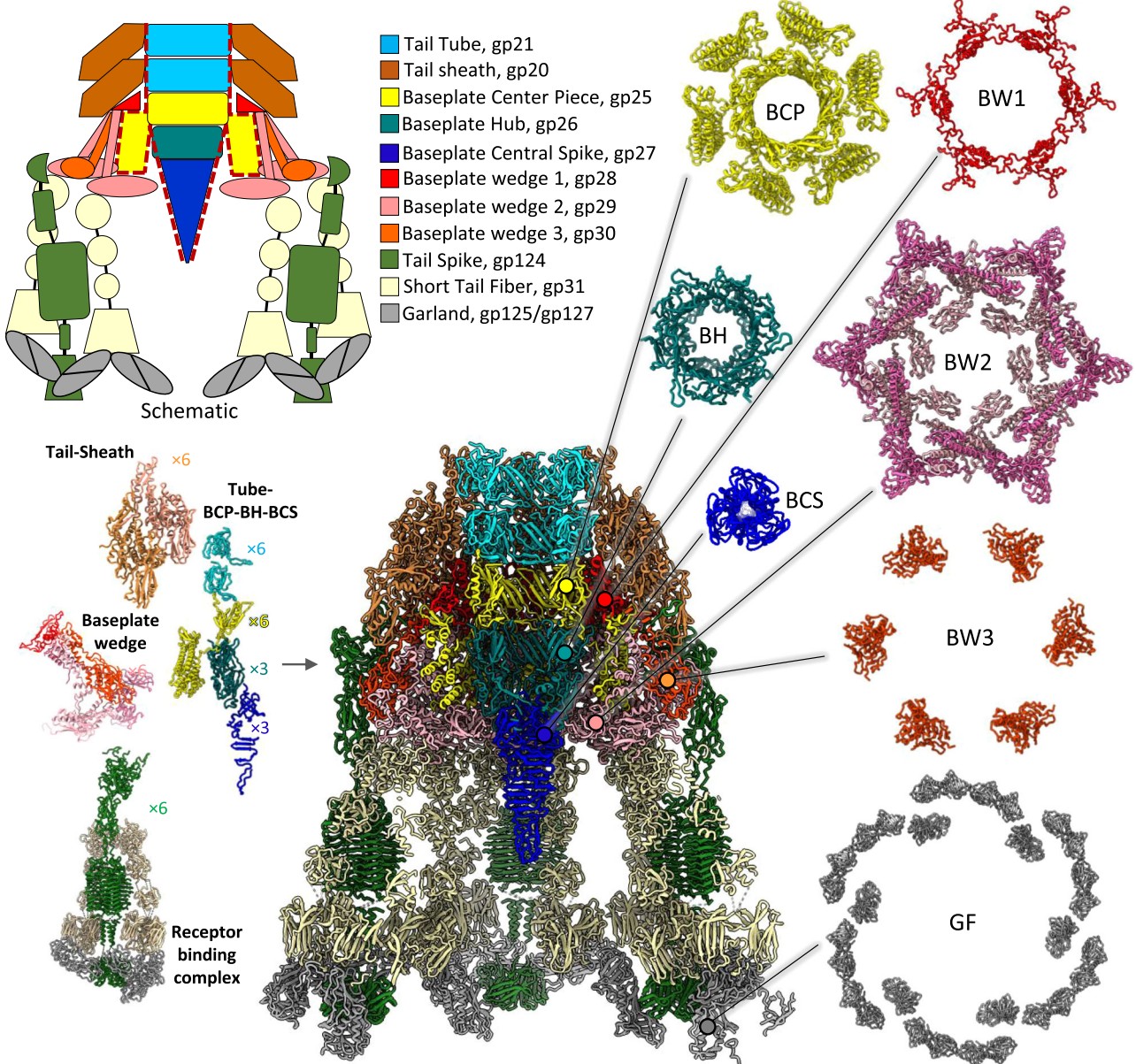

**Fig. 3 | Structural organization of Milano baseplate junction.** A schematic is shown in the upper left corner. The central assembly (tube + baseplate center-piece + baseplate hub + baseplate central spike) is surrounded by a red dashed line. Tail-sheath, tube, baseplate centerpiece (BCP), baseplate wedge (BW), and receptor-binding complex (tail-spike + short tail-fibers + garland) are arranged in C6 symmetry, while baseplate hub (BH) and baseplate central spike (BCS) have a C3 symmetry. Top-view of individual sub-structures (BCP, gp25, baseplate center-piece; BH, gp26, baseplate hub; BCS, gp27, baseplate central spike; BW1, gp28, baseplate wedge 1; BW2, gp29, baseplate wedge 2; BW3, gp30, baseplate wedge 3; GF, gp125/gp127 garland forming proteins) are shown on the right.

introduced previously[23,46] with one modification to account for the new structural information presented here and in other recent reports[25–29]. Namely, the Milano orthologs of bacteriophage Mu Baseplate Hub 1 and Hub 2 proteins will be called baseplate center piece (BCP) and baseplate hub (BH), respectively.

The overall architecture of the Milano baseplate is similar to the minimal baseplate first defined in the study of phage T4 baseplate[26,47] and further refined in the study of R-type pyocin[17]. In addition to the above-mentioned BCP and BH proteins (gp25 and gp26, respectively), the Milano baseplate contains a Baseplate Central Spike (BCS, gp27) and three proteins that form the baseplate wedge complex—Baseplate Wedge 1, 2 and 3 (BW1, BW2 and BW3 or gp28, gp29 and gp30, respectively) (Figs. 3 and 4a, e, f). The small, universally conserved Baseplate Wedge 4 protein, which has a LysM fold (ortholog of T4 gp53 and R-type pyocin PA0627), forms a C-terminal domain of

the Milano BCP (residues 346–396 of gp25) (Fig. 4b, c). In *Photorhabdus* virulence cassette and anti-feeding phage, BW4 is also a C-terminal domain of the BCP, whereas in SPO1-like phages BW4 is fused to BW1 (e.g. in phage A511 BW1 and BW4 are N- and C-terminal domain of gp102)[23].

The structure of the central spike complex of Milano, which consists of BH and BCS proteins, is similar to that of the central spike complex of R-type pyocin (PA0628–PA0616)[17] and phage T4 (gp27–gp5)[48] (Fig. 4d). The BCS contains a double histidine motif in its C-terminal part ([166]HVH[168]), which was shown to coordinate a Fe ion in BCS proteins of phage P2, phi92[49] and Mu[50]. A similar Fe ion is likely present in the Milano BCS because the cryo-EM map of the Milano baseplate shows a spherical peak of 6.5 standard deviations above the mean between the side chains of His166 and His168 matching the expected location of a Fe ion (Fig. 4a). As proposed for other phages,

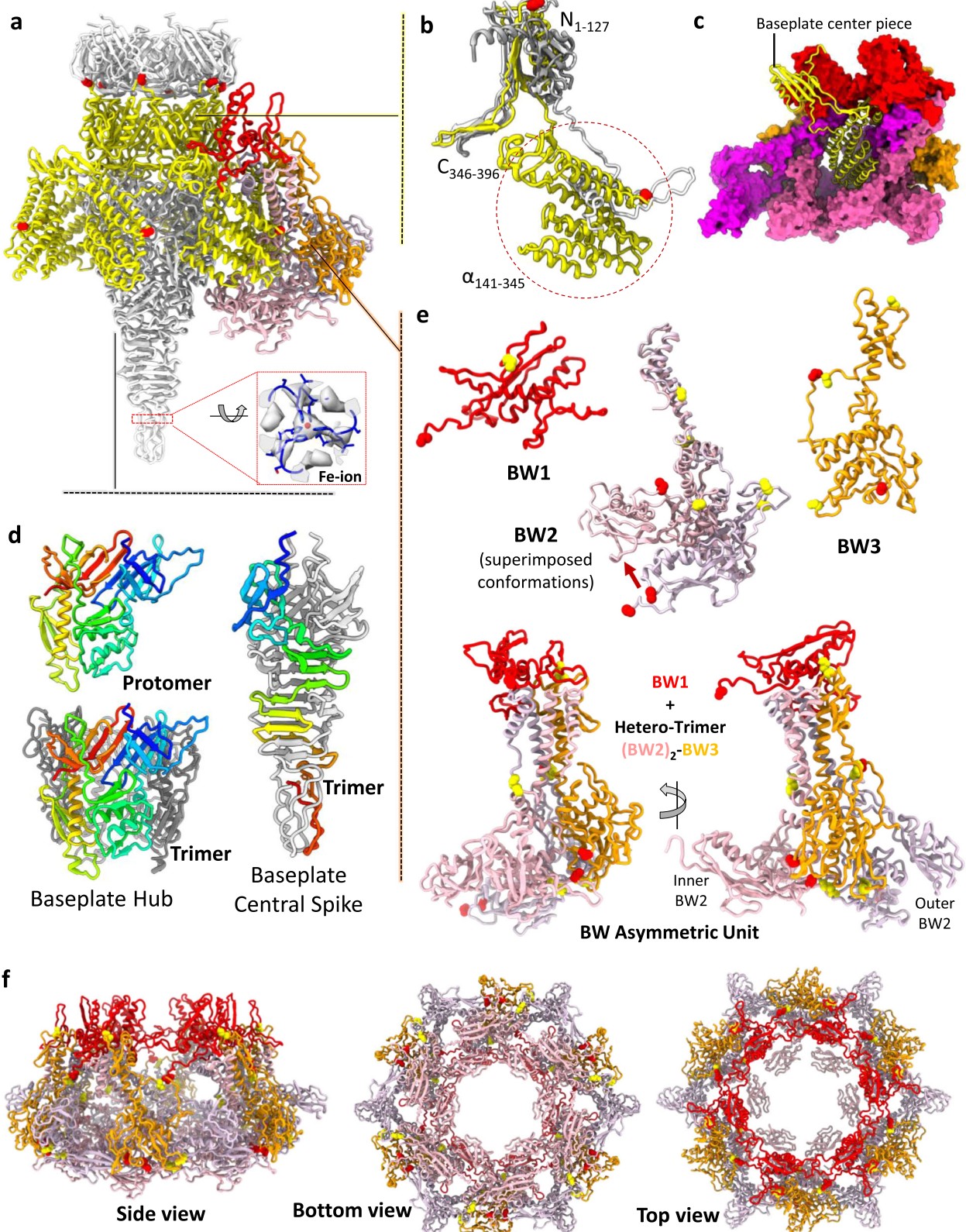

the presence of Fe ion is likely to provide the structural stability to the BCS required for the piercing.

The functions of the three Milano baseplate proteins (BW1, BW2, BW3) plus the C-terminal domain of the BCP (the BW4 domain) are dictated by their position in the baseplate with some novel functional aspects (see below). Similar to the T4 and R-type pyocin baseplate proteins, BW2 (T4 gp6, pyocin PA0618) and

BW3 (T4 gp7, pyocin PA0619) form a (BW2)$_2$–BW3 heterotrimer. Their N-terminal parts of BW2 and BW3 run roughly parallel to each other and create an elongated mostly α-helical structure. Their middle and C-terminal parts fold into globular domains that protrude away from the α-helical structure (Fig. 4e, f). While the N-terminal α-helical part is responsible for giving the baseplate its dome-shaped appearance (i.e. giving the baseplate its "volume"),

**Fig. 4 | Milano baseplate centerpiece (BCP) connects tail, baseplate and central spike. a** Baseplate wedge attached to the central spike assembly. The baseplate centerpiece (BCP, gp25) of Milano is shown in the yellow ribbon model. The tube (gp21), baseplate hub (BH, gp26) and baseplate central spike (BCS, gp27) are colored gray. The baseplate wedge 1 (BW1, gp28), 2 (BW2, gp29) and 3 (BW3, gp30) are colored red, pink, and orange, respectively. The N-terminal tube domain of BCP connects the tail-tube to the baseplate hub. The middle helical domain of BCP is embedded into the groove of the baseplate wedge. **b** Comparison of Milano BCP (yellow) domains with its structural homologs in bacteriophages T4 (gray) and A511 (white). **c** The middle helical domain of Milano BCP (yellow ribbon) is embedded into the groove of the baseplate wedge (surface representation, BW1, red; BW2, pink; BW3, orange). **d** Structure of baseplate hub (BH) and baseplate central spike (BCS). Representative subunit is colored blue-to-red from N- to C-terminus. **e** Structure of BW1, BW2, BW3, and baseplate wedge asymmetric unit (ASU) with highlighted Cys residues involved in intra-ASU (yellow sphere) and inter-ASU (red sphere) disulfide bonds. Two conformations of BW2 protein are shown superimposed. **f** The side, bottom, and top views of baseplate wedge assembly formed by six baseplate wedge ASUs.

the globular domains are responsible for linking the periphery of the wedges together and for the attachment of receptor-binding proteins (RBPs). In previously studied phage baseplates, RBPs bind to BW3. In the Milano baseplate, RBPs bind to the C-terminal domains of both BW2 and BW3 (Figs. 3 and 5).

Generally, in other CIS, BW1 binds to the tip of the $(BW2)_2$–BW3 helical bundle locking the three polypeptide chains together while BW4 binds to the bundle's side. BW4 also interacts with a BW1 belonging to the neighboring bundle (related by six-fold symmetry) in the baseplate. For this reason, BW4 was called a "glue" protein in the pyocin baseplate[17]. The BW1–$(BW2)_2$–BW3–BW4 wedge unit creates a complex surface for binding the first layer of sheath subunits. The Milano BW1 protein, similar to orthologous BW1s, is likely equally important for sheath initiation because its fold is similar to the SHD domain of the sheath. As in all previously described CISs[9,16,17,26,29,51], BW1 accepts the N- and C-terminal arms of the first layer of the sheath subunits, which makes the baseplate an integral part of the sheath by a sheath-like β-sheet augmentation mechanism as described above (Supplementary Fig. 8).

The similarity of the Milano BCP organization to that of the pyocin (PA0626) suggests its central role in the function of the baseplate[17]. In addition to the BW4 C-terminal domain mentioned above (residues 346–396), the Milano BCP protein contains an N-terminal Tube domain (residues 1–127) and a middle Helical domain (residues 141–345) (Fig. 4a–c). The Tube domain forms a smooth continuation of the tube, and its fold and symmetry are both very similar to that of the tube. The Helical domain consists of eight α-helices that are packed into a large bundle, which forms extensive interactions with two neighboring baseplate wedges (Fig. 4a–c). The pyocin BCP protein (previously called Ripcord[17]) contains a similar Helical bundle domain, which is also fused to the C-terminus of a Tube domain. The Helical bundle domain of the pyocin BCP has been shown to serve the wedge-joining, bona fide "glue" function. The post-sheath-contraction pyocin baseplate does not contain the BCP with its helical domain (it is extruded from the baseplate by the moving tube), and the wedges of the pyocin baseplate are splayed out. Most likely, the Milano baseplate undergoes a similar conformational change upon sheath contraction. The poor quality of cryo-EM density for residues 129–139, which link the Tube domain and Helical bundle domain of the Milano BCP, does not allow for building an atomic model. However, this connection is likely in place in the Milano baseplate because the integrity of the polypeptide chain connecting the Tube and Helical domains in the pyocin BCP has been shown to be critically important for pyocin assembly[17].

Similar to the tail-sheath assembly, the Milano baseplate also features an extensive disulfide bond network. Cysteines of BW1, BW2, and BW3 form inter-chain disulfide-bonds within and between baseplate wedges (Fig. 4e, f; Table 1; Supplementary Fig. 3). Furthermore, the Helical domain of BCP is linked by disulfides to BW2 (BCP-Cys318–BW2$_{Inner}$-Cys3) (Fig. 4a–c; Table 1). Considering that the BCP must leave the baseplate to allow the tube to move and sheath to contract and that the wedges of the Milano baseplate might be required to splay out to initiate sheath contraction, the baseplate disulfide bond network must be almost entirely disintegrated prior to sheath contraction. The reducing agent in our in vitro contraction experiments performed this function.

## The structure of Milano RBPs

Although the density for RBP's tail distal domains is not resolved as well as the tail proximal part of the baseplate, it allowed for the high confidence fitting and interpretation of RBP domains and their macromolecular organization (see the "Methods" section). The tail of Milano carries several putative RBPs, which contain domains found in other phage RBPs but are unusual in the way they are attached to the tail and in their overall architecture. As the exact functions of different RBPs are unknown, we will categorize and name them based on their location and structural elements found in other phage RBPs.

A 587 residues long, gp124 is the largest putative RBP. It is a trimeric protein that consists of an N-terminal tail-binding module (residues 1–175), a middle β-helical domain (residues 176–420), and a C-terminal domain (residues 477–579) with a jelly roll β-sandwich fold (Fig. 5a). Such an architecture is typical for a phage tailspike protein (e.g. phage Sf6 tailspike[52]), so we will refer to this protein as Tail Spike Protein or TSP. Neither the β-helical nor C-terminal domain displays strictly conserved residues, so they are unlikely to have an enzymatic activity. There are six TSPs in the Milano tail.

The N-terminal module of the Milano TSP contains two small globular domains (residues 26–120 and 121–175) whose folds are similar to each other and to those of the N-terminal domains of phage CBA120 TSP1[53] and TSP3[54], although the relative orientations and positions of these domains in the Milano spike are different from those of CBA120 TSPs. The span between the Milano N-terminal module and the β-helical domain is formed by a slender fibrous segment consisting of β-hairpins. The β-helical domain is connected to the C-terminal domain by a coiled-coil.

Despite having a typical architecture, the Milano TSP is attached to the tail in an unusual manner (Fig. 5b). As was shown experimentally in phages T4[26], A511[23], in R-type pyocin[17] and as predicted by bioinformatic analysis (e.g. by AlphaFold[55,56]) in many other phages (e.g. P2, Mu), RBPs or RBP complex is attached to the BW3 protein (T4 gp7, A511 gp104, pyocin PA0619, P2 gpI, etc.), which forms the periphery of the baseplate. In contrast, the Milano TSPs are attached to the sheath-baseplate junction region. The 25 residue-long N-terminal arms of the Milano TSP interact with the sheath, the BW1 protein, and the N-terminal part of BW3 "above" the plane of the baseplate. Moreover, the Cys15 residues of the three TSP chains form disulfide bonds with Cys41 of BW1, Cys69 of BW3, or Cys327 of the sheath (the SBD domain) (Fig. 5b). The rest of the N-terminal module of the Milano TSP (the two globular domains), which would be expected to bind tightly to the C-terminal domain of BW3 in a typical baseplate, are "wedged" into a cavity between the middle domain of BW2 and C-terminal domain of BW3. Two proline-rich motifs of the TSP (55-VPPGTP-60 and 100-RTDLPPGPFDPANWQ-115) interact with proline-rich regions of BW2 (residues 150-TDGTLVTPVPGISSAVT-166) and BW3 (155-TYPPSL-160), respectively (Fig. 5b; Supplementary Fig. 9). The significance of this observation remains to be determined.

In addition to the TSP, the Milano tail carries a Short Tail Fiber (STF, 300 residues long), the product of gene 31. Similar to the TSP, the Milano STF is a trimeric protein consisting of a globular N-terminal domain (residues 1–81), which attaches the fiber to the baseplate, a middle slender segment, which consists of two smaller bead-like globular domains (residues 82–120 and 127–160) with similar folds,

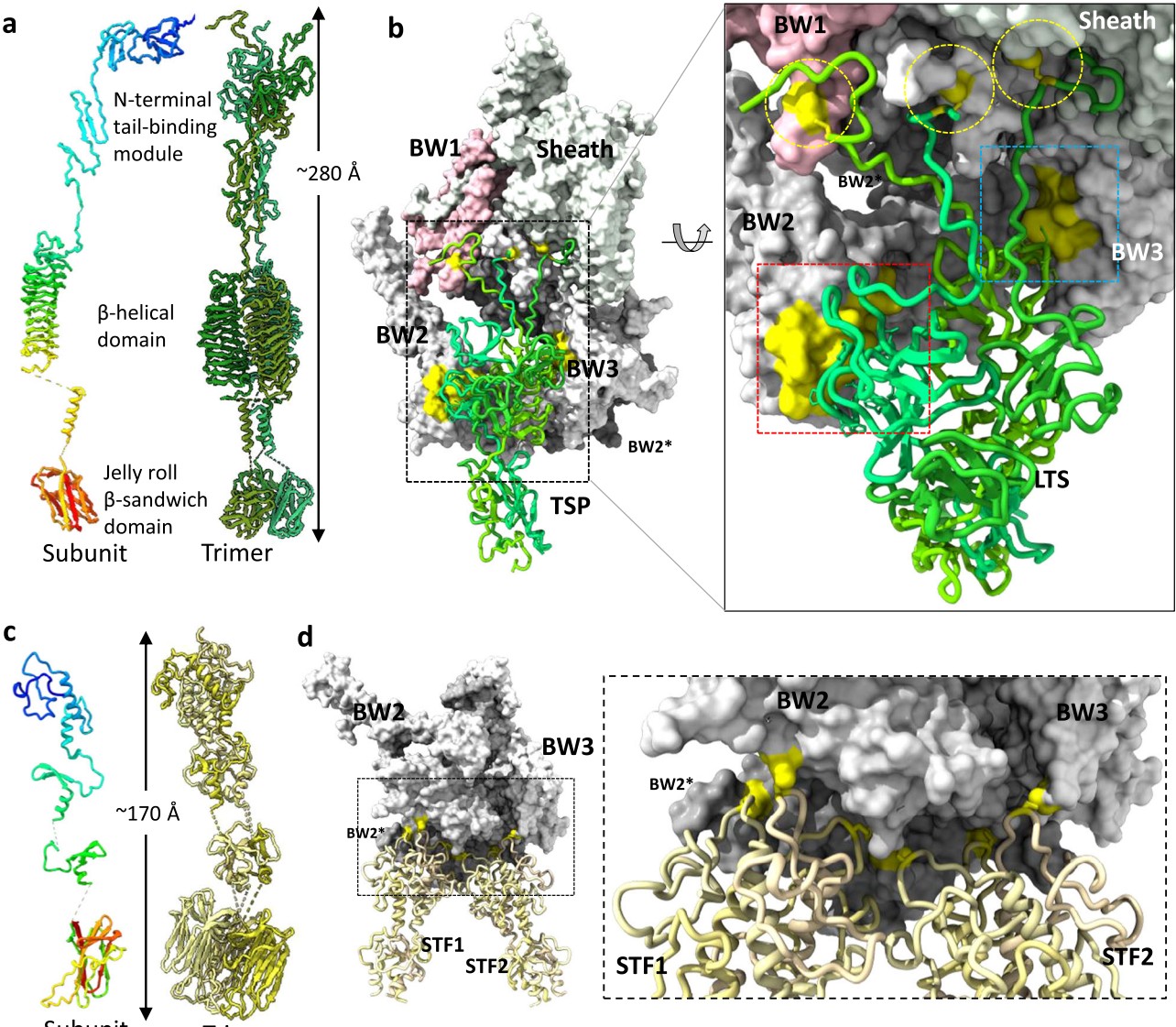

**Fig. 5 | Structural organization of Mialno receptor binding proteins. a** Structure of tail spike (TSP, gp124) subunit and trimer. The N- to C-terminus of the subunit is colored blue-to-red. **b** Interactions of TSP and baseplate components. TSP and baseplate components are shown in the green ribbon model and surface representation, respectively. The TSP interactions with baseplate are highlighted by yellow surface and circles. The cysteine residues involved in disulfide bonds are colored and circled yellow. Detailed protein–protein interactions at red and blue-squared interfaces are shown in Supplementary Fig. 9. **c** Structure of short tail fibers (STF, gp31) subunit and trimer. The N- to C-terminal of subunit is colored blue-to-red. **d** Interaction of STF and baseplate components. STF and baseplate components are shown by tan ribbon model and surface representation, respectively. The cysteine residues involved in disulfide bonds are colored yellow.

and a C-terminal jelly roll β-sandwich domain (residues 169–300) (Fig. 5c). The N-terminal domain and the bead-like domains have novel folds not found in the Protein Data Bank. The N-terminal domain is largely a helical domain made up of six α-helices (two helices per subunit) connected by loops. Middle bead-like domains are formed by stacking of three β-hairpin motifs followed by a short helix of three subunits (Fig. 5c). The C-terminal domain is most similar to a jelly roll found in beta-fructosidase (e.g. PDB id: 1W2T, https://www.rcsb.org/structure/1W2T), but unlike beta-fructosidase, it displays no strictly conserved residues and is unlikely to possess enzymatic activity. The DALI analysis of this domain suggests its similarities to domains of glycosidases, proteases and glycan-binding proteins (Supplementary Table 2).

There are 12 STFs in the Milano tail that could be split into six pairs in which two STFs run roughly parallel to the TSP and interact with it (Supplementary Figs. 9, 10). The strongest interaction is between the C-terminal domains of the STF and the coiled-coil segment of the TSP.

Tails with multiple RBPs are not uncommon (T4, A511, CBA120, G7C, K1–5)[23,26,54,57,58], but in most cases only one RBP binds to the baseplate (in Myophages) or tail (in Podo- and Siphophages), while other RBPs are attached to the first RBP. In the Milano tail, both STFs that surround each TSP are attached to the baseplate directly. Their N-terminal domains bind to the dimeric baseplate circularization module (as defined in ref. 47) formed by two C-terminal domains of adjacent BW2 proteins in the baseplate. Furthermore, all three Cys38 in the STF trimer form disulfide bonds with cysteines in BW2 and, in a single instance, with BW3 (Fig. 5d; Table 1). Here again, the STFs of Milano are attached to the baseplate in a manner not previously described for any phage.

The most unusual part of the Milano RBP ensemble is a garland, which envelopes the distal parts (C-terminal domains) of TSPs and STFs. Its zig-zag pattern resembles the garland of STFs (six gp12 trimers) in the baseplate of phage T4[26] (Fig. 3; Supplementary Figs. 9, 10). In T4, the STF is responsible for the irreversible binding of the phage T4 particle to

the host cell surface[59]. To do so, the garland must unravel to allow the STFs to extend away from the baseplate to reach cell surface receptors. Whether the Milano garland performs a similar function and undergoes a similar transformation remains to be determined.

The quality of cryo-EM density in the garland region does not allow for de novo model building. Compared to all proteins found in the Milano structure (Supplementary Table 3) AlphaFold models of homo- and heterodimers of gp125 and gp127 are predicted to be very reliable (average pLDDTs of 90), the two models are virtually identical (the proteins show ~30% sequence identity), and fit the cryo-EM map equally well (Supplementary Fig. 10). Thus, the garland contains 18 homo- and/or hetero-dimers of gp125 and gp127 (36 polypeptide chains) with three dimers associated with each TSP–STF–STF complex. There are also six additional blobs of weak cryo-EM density at the periphery of the garland that cannot be reliably fitted with any of the Milano proteome AlphaFold models. Taking into account that RBP genes often form contiguous clusters, the extra density might correspond to a globular domain of a gp126 monomer. However, the latter has not been detected in the proteome. Of note, neither gp125 nor gp127 contain cysteines, so there are no disulfide bonds in the garland.

The second peculiarity of the garland proteins is that their fold is similar to that of T4 gp8[60] and A511 gp105[23]. Both form dimers that are similar to each other and to the dimers of Milano gp125 and/or gp127. Unlike the Milano garland proteins, these proteins are located near the N-terminal ends of RBPs, i.e. close to the baseplate. They play an important role in connecting RBPs to the baseplate and were proposed to be called Baseplate Wedge protein 3−Tail Fiber Network connectors (BW3TFN)[23]. Notably, in the A511 baseplate, three BW3TFN dimers form a chain, like beads on a string, in which an extended chain of the BW3 protein runs across each BW3TFN dimer. Moreover, when recombinantly produced and crystallized, these proteins form a nearly identical beads-on-a-string structure in crystallo by donating and accepting each other's His-tags. The Milano garland proteins gp125 and gp127 appear to form a similar filament, but it is unclear what serves as a "glue" or "string" here. Gp126 would be a possible candidate as its AlphaFold model contains a long N-terminal string.

## Discussion

Our atomic structures of the Milano tail in both the extended and contracted states reveal the structural origin of tail flexibility and help to understand how a flexible tail can transform into a rigid one upon contraction without structural rearrangement in the tail tube. Our structures also provide confirmatory evidence about the absence of conformational changes in the tube upon contraction, something that has only been hypothesized previously[59].

The Milano tail tube protein is flexible due to the missing loop between residues 111 and 112 (Fig. 1E; Supplementary Fig. 2c), which is present in tubes of other CISs. In this respect, the Milano tube resembles the tail tube protein of various Siphophages, which further supports a hypothesis that Myophage tails evolved from Siphophage tails by acquiring a sheath[61]. Taking into account the reduced fraction by which the Milano sheath contracts compared to other sheaths, Milano-like phages might represent an evolutionary ancient clade of Myophages in which the sheath contraction apparatus has not been fully developed.

Grooves on the tube and electrostatic potentials of tube/sheath surfaces provide the Milano tail with a dual physical nature. The Milano tail is flexible enough (like non-contractile tails) to resist strong mechanical forces in the extended state while rigid in the contracted state, allowing it to pierce the cell membrane. We think that the flexible-to-rigid transformation of the tube upon contraction may also be partly driven through the removal of the tape measure protein (TMP) from the lumen of the tube. The luminal surface of the Milano tube is negatively charged facilitating DNA transport (Supplementary Fig. 2f). The negative electrostatic potential on the tube's luminal surface can act in a similar way to the outer surface, being masked by the TMP while extended and contributing to the tube stiffness in the absence of TMP upon contraction.

We believe that the flexible nature of the Milano tail in the extended state is an adaptation to cope with a large mechanical stress resulting from binding to a rapidly rotating flagellar filament. A flexible and bendable tail can prevent the premature trigger of contraction, an unproductive release of the genome, or damage to the phage particle, which could occur under mechanical stress. While it is sometimes believed that a rigid structure may be stronger than a flexible one, the opposite is usually the case where rigid structures can be quite fragile, while flexible ones can have the elasticity to tolerate large forces. Tobacco mosaic virus is a good example of a rigid structure that can be easily fragmented by shear forces[62]. In contrast, the flexible Type 1 pili of Uropathogenic E. coli can withstand the very high shear forces present in the urinary tract[63,64]. Similarly, the flexible tail of the YSD1 phage is proposed to sustain higher mechanical stress[51].

On average, Milano proteins contain 1.4% cysteine residues. This fraction is higher than that found in other phages (e.g. 0.9% in T4 and 0.7% in A511). Cysteines are even more abundant (2.1%) in Milano's structural proteins. Many of these cysteines form intra- (Supplementary Table 4) and interchain disulfide bonds (Table 1), which, on the one hand, play a role in the structural stability of the particle, but, on the other, must be broken to enable tail contraction, which is a prerequisite for DNA delivery into a host cell. Although this is beyond the scope of the present structural study, we can speculate that the reduction of disulfide bonds in the Milano sheath and baseplate during natural infection may be due to one or more of the following factors: (i) as a result of high growth rate, the host depletes the local environment of oxygen; (ii) the local environment undergoes acidification; and (iii) a specific reducing agent is present on the surface of the host cell.

The manner by which putative RBPs of Milano are attached to the baseplate has not been described previously in any other phage. The largest RBP (a TSP) interacts with the sheath protein directly while two shorter RBPs (two identical STFs) bind to the BW2 protein. All three RBPs—the TSP and both STFs—form disulfides with various baseplate proteins and, remarkably, the TSP is linked by disulfides to the sheath protein. These bonds likely permit the system to better coordinate host cell surface recognition with sheath contraction. However, it is unclear whether these disulfides would be preserved in an environment in which other disulfides are reduced, as required for contraction.

The Milano RBP complex is quite different from that of flagellotropic Siphophages. The single thread-like tail fiber of Siphophages Chi and YSD1, which is likely polymerized from one individual protein species, wraps around the host's flagellar filament. The Milano RBP complex contains no such fiber but includes at least four different types of proteins—TSP (gp124), STF (gp31), and garland (gp125 and gp127) that are connected to each other. Translocation of Chi/YSD1 along the flagellar filament has been proposed to follow a "nut-and-bolt" mechanism[51]. Due to the relatively small size and inter-connected nature of Milano receptor binding proteins, it is unlikely to follow a similar mechanism of translocation and it will be the subject of further research.

## Methods

### Cultivation and purification of Milano particles

Bacteriophage Milano was propagated using an overlay plate method. *Agrobacterium tumefaciens* C58 was grown in TYC broth (5 g/L Bacto tryptone, 3 g/L (yeast extract, 0.86 g/L CaCl$_2$·2H$_2$O) shaking vigorously at 30 °C to an OD 600 of 0.6. Motility was verified by phase contrast microscopy. Next, 100 mL of motile culture was combined with ~10$^6$ plaque-forming units of bacteriophage Milano. After a 5-min incubation at room temperature, 4 mL of molten 0.35% TYC soft agar was added to the culture−phage mixture, which was immediately poured

onto a TYC 1.5% agar plate. This procedure was repeated for a total of 20 plates, which were then incubated at 30 °C overnight until lysis was visible on the plates. A total of 5 mL of TM buffer (50 mM Tris–HCl, pH 7.4 and 10 mM MgSO$_4$) was gently pipetted onto each plate. Plates were placed face-up on a gyrating platform at 4 °C for 24 h, rotating slowly at 60 RPM. Liquid containing buffer and loose soft agar from all plates was pooled together in a 250 mL centrifuge bottle and combined with agar lawns that were scraped into the bottle. After the addition of 1 mL chloroform, the agar/buffer mixture was shaken vigorously for one minute to liberate phage particles. The mixture was centrifuged at 10,000 × g for 30 min at 4 °C, then the supernatant was carefully pipetted into another centrifuge bottle. To precipitate phage, NaCl and polyethylene glycol 8000 were added to the lysate to a final concentration of 1 M and 10% (w/v), respectively. The precipitating lysate was mixed on a magnetic stir plate at slow rotation for 24 h at 4 °C. The precipitated lysate was centrifuged at 15,000 × g for 30 min at 4 °C, and the supernatant was discarded. The phage-containing pellet was suspended in 2 mL of cold TM buffer. A linear 10–50% OptiPrep™ density gradient was prepared in disposable ultracentrifuge tubes, the phage lysate was layered carefully on top of the density gradient and centrifuged at 200,000 × g in a Beckman Coulter Optima™ L-90K ultracentrifuge for 2 h at 15 °C using the SW41 swinging bucket ultracentrifuge rotor (Beckman Coulter, USA). After centrifugation, a cloudy blue band appeared within the density gradient, which was extracted using a syringe and 18-gauge needle, transferred to a 10,000 MWCO Thermo Scientific Slide-A-Lyzer® dialysis cassette, and dialyzed against TM buffer for 3 days at 4 °C, changing the buffer once per day. Purified phage was extracted from the cassette and titered via plaque assay.

## Spot assay

*Agrobacterium tumefaciens* strain C58 and its derivatives, BM140 (DflaA-D)—flagella-minus (fla⁻) strain[65] and PMM5 (DmotA)—the non-motile (mot⁻) strain[66] were grown in 3 mL of TYC (0.5% tryptone, 0.3% yeast extract, 0.087% CaCl$_2$·2H$_2$O) supplemented with streptomycin for 16 h at 30 °C. Each culture was diluted 30-fold with fresh medium and grown under shaking conditions for 5 h at 30 °C to an OD600 of 0.6. The wild-type strain was checked microscopically for motility. One hundred microliters of each culture were mixed with 3.5 mL of molten 0.35% TYC soft agar, poured onto pre-warmed TYC 1.5% agar plates supplemented with streptomycin, and allowed to solidify. Serial dilutions of Milano phage in TM buffer (50 mM Tris–HCl, pH 7.4 and 10 mM MgSO$_4$) were prepared from a concentrated stock at $1.85 \times 10^{11}$ pfu/mL. Two microliters of phage stock, serial dilutions up to $10^{-14}$, and a buffer control were spotted on the Agrobacterium overlay plates in horizontal rows. Plates were incubated overnight at 30 °C.

## Induction of contraction

Phage particles were incubated with different combinations of chaotropic chemicals and reducing agents: (1) 3–7 M Urea, (2) 3–7 M guanidine HCl, (3) 3 M Urea + 5 mM DTT, (4) 3 M guanidine HCl + 5 mM DTT, and (5) 5 mM DTT, in phage storage buffer (50 mM Tris–HCl, 10 mM MgSO$_4$, pH 7.0) for 30 min at room temperature. Treated phage particles were centrifuged at 120,000 × g for 90 min at 4 °C and resuspended in a storage buffer before microscopy. Contraction was successfully induced with 3 M guanidine HCl and 5 mM DTT.

## Negative staining electron microscopy (NS-TEM)

Three microliter of phage sample was applied to glow-discharged carbon film grids (300 mesh, copper), negatively stained with 2% uranyl acetate, and imaged by a Tecnai T12 transmission electron microscope using standard settings. NS-TEM micrographs were used to interpret the morphology (curved/straight, normal/contracted) of the phage tail.

## Cryo-electron microscopy

**Sample preparation and image preprocessing.** A 3 μL sample (either extended or contracted phages) was applied to glow-discharged holey C-flat carbon grids (1.2/1.3, 400 mesh, copper) and then plunge-frozen using an EM GP Plunge Freezer (Leica). Cryo-EMs datasets for normal and contracted phage samples were collected on a 300 keV Titan Krios with a K3 camera (University of Virginia) at pixel size 1.08 and 0.82 Å, respectively, with a total dose of ~50 e Å$^{-2}$. Micrographs were motion corrected and CTF (contrast transfer function) was estimated by 'patch motion correction' and 'patch CTF estimation' jobs in cryoSPARC, respectively[27,67,68].

**Reconstruction of curved-tail.** The 3D reconstruction of curved-tail was achieved using cryoSPARC[68] as described previously[24]. Particles from the tail were auto-picked by 'Filament Tracer'. The picked particles were 2D classified and classes with bad particles were removed. The selected particles were used initially for helical reconstruction to assign alignment angles to particles and generate a mask. The aligned particles from this job were clustered by '3D variability analysis (3DVA)'. The particle cluster with the largest number of particles (~228,393 particles) and visible curvature was subjected to reconstruction with no imposed symmetry by multiple iterations between 'Local refinement' and 'Local CTF refinement' until the resolution stopped improving.

**Single particle reconstruction of baseplate junction.** The 3D reconstruction of the baseplate junction was performed by a standard single particle analysis pipeline with C3 symmetry in cryoSPARC[68]. Particles for baseplate junction were picked manually to get 2D class averages later used as templates for final particle picking by 'Template picker'. Bad particles were removed by 2D-classification. Selected particles were subjected to 'ab-initio reconstruction' to generate the low-resolution volume. The low-resolution model was further refined by iterative cycles of 'nonuniform homogenous refinement' and 'local CTF refinement' jobs with C6- and C3-symmetry. The map with C6 symmetry showed the blurred density in the region corresponds to the baseplate hub and central spike. The C3-map shows reasonably resolved density for this region and was used for model building.

**Helical reconstruction of contracted tail-sheath and tail-tube.** For the sheath, particles were auto-picked by 'Filament tracer' and subjected to 2D classification to remove bad particles. Selected particles were used to generate an averaged power spectrum. Potential helical symmetries were calculated from the power spectrum and tested by 'Helical refinement' with a small subset of particles to find the correct symmetry. Initial helical symmetry parameters and the resolution of the map were further refined by iterative cycles of 'Helical refinement' and 'Local CTF refinement' jobs.

For the tube, we first tried to pick the extruded part of the tube, which was not sufficient. Therefore, we used the tube surrounded by the contracted sheath. The particle stack used for the final sheath reconstruction was used for tube reconstruction, too. Initially, the tube symmetry-indexing and reconstruction were hindered by the dominant signal from the contracted sheath having a different symmetry. The particles were windowed to only include the central part of the image containing the tube. The averaged power spectrum of such windowed particles allowed for the correct indexing of the tube's helical symmetry. For reconstruction of the tube, every step of 'Helical refinement' and 'Local CTF refinement' was provided with a manually created static 3D mask excluding the sheath region to avoid misalignment due to signal from the sheath. The statistics of cryo-EM data collection and processing are detailed in Supplementary Table 5. The density map, model-to-map fitting, and map-to-map Fourier shell correlation curves for all reconstructions are shown in Supplementary Fig. 11.

## Model building

All 127 proteins of Milano were predicted by AlphaFold[69] using the ColabFold server[70]. The fitting of predicted protein structures into 3D maps was performed both manually and by DeepTracer-ID, as needed[71]. Once a protein component of the complex map was identified, the fitted AlphaFold model was refined against the density map by iterative cycles of interactive refinement in Coot[72] and real-space refinement in PHENIX[73]. The density for TSP residues 175–587, STF residues 127–300, and garland proteins are not resolved and thus their AlphaFold predicted models were only rigid body fitted and not included in the protein data bank deposition, 8FQC. The combined model of baseplate and full receptor binding proteins is provided as Supplementary Data 1 and used for figure preparation and analysis. The quality of baseplate structure map allowed the identification of a protein gp28, which has not been annotated due to an error in DNA contig assembly (personal communication with author of ref. 1). The structure of the straight tail was extracted from the baseplate junction structure. The statistics of model building are detailed in Supplementary Table 5.

## Mass spectrometry

**Sample digestion.** The samples were prepared similar to as described[74]. Briefly, 42 μL of phage sample is mixed with 42 μL of 10% SDS, 100 mM Bis−tris propane, pH 6.5. The sample was alkylated with 8.4 μL of 44 mM N-ethylmaleimide and allowed to react for 3 h at 37 °C in the dark. 4.7 μL of 12% phosphoric acid was added to the 92.4 μL protein solution followed by the addition of 294 μL of binding buffer (90% methanol, 100 mM Bis−tris propane pH 6.5). The resulting solution was added to S-Trap spin column (protifi.com) and passed through the column using a benchtop centrifuge (60 s spin at $1000 \times g$). The spin column was washed with 150 μL of binding buffer and centrifuged and repeated two times. 30 μL containing 600 ng of Chymotrypsin in 50 mM Bis−tris propane pH 6.5 was then added to the protein mixture and incubated at 37 °C for 3 h. 30 μL containing 600 ng of Trypsin in 50 mM Bis−tris propane pH 6.5 was then added, and incubated at 37 °C overnight. Peptides were sequentially eluted with 30 μL of water, 30 μL of 0.2% formic acid, and finally three elutions with 30 μL of 70% acetonitrile, 0.1% formic acid. The combined peptide solution is then dried in a speed vac and resuspended in 1.67% acetonitrile, 0.08% formic acid, 0.83% acetic acid, and 97.42% water and placed in an autosampler vial.

**NanoLC MS/MS analysis.** Peptide mixtures were analyzed by nanoflow liquid chromatography–tandem mass spectrometry (nanoLC–MS/MS) using a nano-LC chromatography system (UltiMate 3000 RSLCnano, Dionex), coupled on-line to a Thermo Orbitrap Fusion mass spectrometer (Thermo Fisher Scientific, San Jose, CA) through a nanospray ion source (Thermo Scientific). A trap and elute method was used. The trap column was a C18 PepMap100 (300 μm × 5 mm, 5 μm particle size) from Thermo Scientific. The analytical column was an Acclaim PepMap 100 (75 μm × 25 cm) from (Thermo Scientific). After equilibrating the column in 98% solvent A (0.1% formic acid in water), and 2% solvent B (0.1% formic acid in acetonitrile (ACN)), 8 μL of sample was injected onto the trap column and subsequently eluted (300 nL/min) by gradient elution onto the C18 column as follows: isocratic at 2% B, 0–5 min; 2–4% B, 5–6 min; 4–25% B, 6–95 min; 25–50% B, 95–120 min; 50–90% B, 120–125 min; 90% B for 1 min, 90% to 4% B 126–129 min; 4–90% B from 129 to 132 min; 90% to 2%, 135–137 min; and isocratic at 2% B, till 150 min.

All LC−MS/MS data were acquired using XCalibur, version 4.4 (Thermo Fisher Scientific) in positive ion mode using a data-dependent acquisition (DDA) method with a 5 s cycle time. The survey scans ($m/z$ 375–2000) were acquired in the Orbitrap at 120,000 resolution (at $m/z = 400$) in profile mode, with a maximum injection time of 50 ms and an AGC target of 400,000 ions. The S-lens RF level was set to 60. Isolation was performed in the quadrupole with a 1.6 Da isolation window, and HCD MS/MS acquisition was performed in centroid mode with detection in the Orbitrap at 30,000 resolution, with the following settings: parent threshold = 25,000; stepped collision energies = 20%, 30%, 40%; maximum injection time 100 ms; AGC target 25,000 ions. Each HCD MS/MS acquisition was followed by an ETD MS/MS acquisition performed in centroid mode with detection in the Orbitrap at 30,000 resolution with the following settings: parent threshold = 25,000; maximum injection time 250 ms; AGC target 100,000 ions, and use calibrated charge dependent ETD parameters. Monoisotopic precursor selection (MIPS) and charge state filtering were on, with charge states 2–10 included. Dynamic exclusion was used to remove selected precursor ions, with a ±10 ppm mass tolerance, for 30 s after the acquisition of one MS/MS spectrum.

**Database searching.** Tandem mass spectra were extracted and the charge state was deconvoluted by Proteome Discoverer (Thermo Fisher, version 2.5.0.402). Deisotoping was not performed. All MS/MS spectra were searched against the UniProt database of *Agrobacterium* phage Milano and common laboratory contaminants (cRAP, thegpm.org) using Sequest. Searches were performed with a parent ion tolerance of 10 ppm and a fragment ion tolerance of 0.02 Da. Trypsin/Chymo is specified as the enzyme (fully specific to cleavages at RKFYWL), allowing for four missed cleavages. Variable modifications of N-ethylmaleimide (C) and oxidation (M) were specified in Sequest. Protein identifications were exported as a focused protein FASTA database. Disulfide cross-linking sites were identified from the LC-MS/MS data using pLink2 (Version 2.3.9, pFind Team, Beijing, China)[75] using the pLink-SS workflow against the created focused protein FASTA database. pLink2 search parameters were as follows: enzyme non-specific, missed cleavages 3, peptide mass 400–6000 Da, peptide length 4–60, precursor tolerance 20 ppm, fragment tolerance 20 ppm. The results were filtered at a 5% FDR cutoff, E-value < 0.001. Results from pLink2 were summarized using the Python script provided by Lu et al. [76]. Disulfide cross-linking sites were inspected and validated manually.

## Structural and sequence analysis

All structures were analyzed and displayed by ChimeraX[77]. Protein structural and sequence similarity searches were performed by DALI[78] and NCBI-BLAST (https://blast.ncbi.nlm.nih.gov/Blast.cgi), respectively. The electrostatic potential of protein surfaces was calculated by APBS−adaptive Poisson−Boltzmann solver[79].

## Reporting summary

Further information on research design is available in the Nature Portfolio Reporting Summary linked to this article.

## Data availability

The atomic models and three-dimensional reconstructions discussed in the article are available in the Protein Data Bank (IDs: 8FOP, 8FQC, 8FOU, and 8FOY) and Electron Microscopic Data Bank (IDs: EMD-29353, EMD-29383, EMD-29354, and EMD-29355), respectively.

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

## Acknowledgements

This work was supported by NIH GM122510 (to E.H.E.). B.E.S. is supported by National Science Foundation fund number IOS-2054392. P.G.L. is supported by NIH GM139034. Electron Microscopy was performed at the University of Virginia Molecular Electron Microscopy Core facility. The mass spectrometry was performed at the Mass Spectrometry Facility (at UTMB-Health) supported in part by the Cancer Prevention Research Institute of Texas (CPRIT) grant number RP190682. We thank Clay Fuqua for providing the *Agrobacterium tumefaciens* strains used in this study. We thank Tomek Osinski for computational support.

## Author contributions

R.R.S., E.H.E. and B.E.S. designed the study, while P.G.L. provided additional suggestions for experiments. The phage sample preparation and spot assay were conducted by N.C.E., A.A.H., A.L.S. and R.J.K. under the guidance of B.E.S. Mass spectrometry and disulfide mapping was performed by L.K.P. and W.R. Cryo-EM grid preparation, experiment, image-analysis, reconstructions, and model building were conducted by R.R.S., with suggestions from E.H.E., P.G.L., F.W. and M.A.B.K. The phage contraction experiment was performed by R.R.S. with suggestions from P.G.L. and E.H.E. Manuscript and revisions were prepared by R.R.S., E.H.E., P.G.L. and B.E.S. Additional suggestions on the manuscript were provided by F.W. and M.A.B.K.

## Competing interests

The authors declare no competing interests.
