## [Peer Review File · Nature Communications]

REVIEWER COMMENTS

Reviewer #1 (Remarks to the Author):

This work reports the first structure of a flexible, contracting tails. This is a fascinating system that departs from "classical" models in many ways. This structural study will be of great interest to the field and opens interesting research avenues that should be fruitful in coming years.

Specifically, the manuscript describes the pre- and post-conformation of the tail tube and sheath components of the tail, providing an experimental validation of a long-held hypothesis that the tail-tube is unchanged in the process. This was far from evident, especially for a tail that undergoes a transition from bent to straight. The structure also supports and adds to previous hypotheses made for Siphoviridae about the molecular basis for the tail plasticity.

In contrary, the study reveals surprising findings in the architecture of the baseplate and the presence of disulfide bonds that lock the structure in a close state and suggests a redox trigger for entry.

I very much enjoyed reading this well-written manuscript and discovering all these intriguing features. I strongly believe it will be of major interest to the field of phage research and more broadly virology.

Here are my detailed comments to clarify some points and improve readability:

p.4 "It oligomerizes by extending the loop formed by residues 44-61 into the neighboring subunit in the hexamer."

In other phage tail tubes, the oligomerisation involves a continuous beta-barrel and the long loop contributes to inter-ring contacts. From the figures, it seems to be the case for the Milano phage too but the description doesn't match.

p.5: "However, the loop, which is absent in the Milano tube (between residues 111-112), is present in these phages, suggesting that the structural origin of grooves in these bacteriophages is different from that of Milano."

Could this be illustrated in supplementary material? It is not clear to me that this loop is present and participating to inter-ring contacts in YSD1, 80alpha or SPP1. In YSD1 which has the most pronounced groove (Fig. S2), removal of the long loop abolishes almost all inter-ring contacts, is this the case here too?

p.6/7: where are the 10-12 residues considered as "non-equivalent" in the superimposition located? Is this a single loop or are they scattered across the tail tube protein?

p.8 induction of tail contraction by guanidine and DTT:

Only one small image of the straight tail is presented. More should be provided in supplementary material to allow the reader to see how straight they are in general and whether the protruding part of the tube is indeed straight without the external sheath. This is strongly suggested by the existing image but it is too small and isolated to be definitive.

p.8: "These cysteine residues are conserved in other Agrobacterium-infecting phages, such as 7-7-13 (Uniprotkb, J7FA80) and OLIVR4 (Uniprotkb, A0A858MST0)."

Is there a broader picture of when/where these disulfide bond locks emerged? These are a fascinating find and beg the question of how their release is mediated in vivo. Is this something specific to Agrobacterium or more general?

In Fig. 2, I couldn't quite follow the separation of the two "locking" cysteines and the disruption of

the disulfide bond during contraction. It may be useful to change the color coding for these specific residues (Cys 47 and 182) so they can be tracked across the different panels (i.e. pre- and post-contraction).

Fig. 4E: BW3 should be in the same orange shade as 4F for readability. It may be clearer if the disulfide bonds between heterotrimers are colored differently from the intra-trimer disulfide bonds. Lettering of the panel is incorrect.

p. 10: "As in all previously described CISs14,39,43-46, BW1 accepts the N- and C-terminal arms of the first layer of the sheath subunits, which makes the baseplate an integral part of the sheath by a sheath-like β -sheet augmentation mechanism as described above."

This is not evident in the presented figures.

Given that the short-tail fiber protein displays 2 novel domains, these should be described in more details. A description of the folds and a Jones-rainbow representation would be useful. When structural comparison is mentioned (p. 12 last paragraph), it would also be important to have the key results from DALI and PDBefold searches (e.g. Z-scores, length of alignment, rmsd etc.) for the top hits even if somewhat distant.

p.10: "The post-sheath-contraction pyocin baseplate does not contain the BCP with its helical domain (it is extruded from the baseplate by the moving tube), and the wedges of the pyocin baseplate are splayed out. Most likely, the Milano baseplate undergoes a similar conformational change upon sheath contraction."

Could the authors describe a bit more what is observed for the BP in contracted tails? Even if reconstructions have been impossible so far, there may be observations from the raw cryoEM images / negative stain.

Does the garland need to be dissociated to give way to the tail tube?

Discussion:

"We believe that the flexible nature of the Milano tail in the extended state is an adaptation to cope with a large mechanical stress resulting from binding to a rapidly rotating flagellar filament."

To add support to this proposal (and the following paragraph), it may be useful to mention that a similar hypothesis about mechanical stress and plasticity has been proposed for YSD1, which is also a flagellotropic phage.

Reviewer #2 (Remarks to the Author):

The paper by Sonani et al. describes an elegant structural analysis of the *Agrobacterium tumefaciens* bacteriophage Milano. The authors determined a cryo-EM reconstruction of the contractile tail of phage Milano in the bent and contracted conformations, deciphering the bent-to-straight transformation of the sheath and tube proteins. Unexpectedly, the authors found that the Milano tail sheath and baseplate are covalently linked by multiple disulfide bounds, which may enable the phage to survive the mechanical stress by binding to the host flagella. Overall, these findings broaden our understanding of covalent crosslinking in bacteriophages and shed light on the biology of flagellotropic phages that do not carry a curly fiber to wrap around the host flagellum.

Overall, I believe this paper has many strengths and advances the field of bacteriophage biology, especially for flagellotropic phages that are poorly understood. At the same time, there are several gaps that the authors should address to make this work publication quality:

1. Introduction. Can the author clarify the evidence that Milano is a flagellotropic phage like 7-7-1? Reference 2 is quite old (1977). Is Milano inferred to be a flagellotropic phage based on sequence similarity to 7-7-1, or was this demonstrated experimentally? In general, I believe the introduction of this paper should focus more on Milano rather than generalizing Milano and 7-7-1 as if they were the same phage.

2. Page 4. End of Introduction. By all means, tone down the claim that this paper is the first experimental proof that the tail tube remains in the same conformation after contraction. Several published papers reached the same conclusion (for non-flagellotropic phages/pyocins):

- Feng Jian et. al. 2019. Cell. Cryo-EM Structure and Assembly of an Extracellular Contractile Injection System.
- Desfosses et. al. 2019. Nat Microbiol. Atomic structures of an entire contractile injection system in both the extended and contracted states.
- Peng Ge et. al. 2020. Nature. Action of a minimal contractile bactericidal nanomachine.
- Brenda Gonzalez et. al. 2021. Viruses. Structural Studies of the Phage G Tail Demonstrate an Atypical Tail Contraction.

3. Introduction and Discussion. Milano is said to be flexible and bent, but a bent tail does not imply flexibility. A boomerang is bent yet very sturdy. Can the authors clarify what biochemical/biophysical features/evidence support the claim that Milano's tail is flexible?

4. Figure 4 is overwhelming. Can it be simplified to look more like Figure 3?

5. Induction of contraction. The authors use 5 mM DTT plus 3M urea + 3 M guanidine HCl for 30 min to induce contraction. What happens if the denaturants (3M urea + 3 M guanidine HCl) are taken out of the reaction? How can they pinpoint the specificity of contraction to just DTT? Is DTT necessary and sufficient for tail contraction, or just necessary? Similarly, would a higher concentration of denaturants without DTT induce contraction?

6. Did DTT and denaturants result in complete genome ejection? Did the authors observe virions with a contracted sheath and genome inside the head? Can this point be clarified and quantified accurately?

7. How were disulfide bonds identified, and what 'hard' evidence supports their existence? Were they identified based on the electron density? Can an alternative method be used to demonstrate that Milano's structural proteins are cross-linked? For instance, native gel, SDS-PAGE, or, better, mass spec analysis. I am positive the authors have seen the paper from the Zhou's lab on phage Pam3 that just came out in PANS, where MS was utilized
<https://www.pnas.org/doi/10.1073/pnas.2213727120>

8. Pages 8-11. The authors provide an extensive description of the baseplate in the extended state. It would also be helpful to present the structure of the contracted baseplate. Was it observed at all? Are there significant structural differences between the two conformations?

9. Can the authors speculate on the biology of *Agrobacterium tumefaciens* infection? When/where would the phage Milano encounter a different redox environment to trigger disulfide bond reduction/oxidation?

10. Discussion. The authors mentioned the tape measure protein. Have they found any evidence for the tape measure protein in their reconstruction? Can this be shown/discussed?

11. Suppl Fig 7. The reconstructions obtained without (and to some extent with) a mask reveal strikingly poor correlation at the medium-to-low resolution, typically indicative of multiple conformations. This can be observed in poor(ish) electron densities, despite the FSC resolution at 0.143. Can the authors comment on this point? Can the experimental density be shown in the suppl. Info? In my experience, densities like this do not refine well against atomic models using real-space refinement (see next point)

12. Table 2. The authors must report the model-to-map CC, which is our best surrogate for the Rfree used in crystallography. At this resolution, all models presented in this paper and deposited in the PDB are expected to have CC>0.75. This is a vital point.

Reviewer #3 (Remarks to the Author):

The manuscript entitled "Agrobacterium tumefaciens phage Milano requires a redox trigger for tail contraction" by Sonani and co-authors has been submitted to the Nature Communications journal. The phage Milano represents a group of flagella-tropic phages and infects hosts through binding an actively rotating flagellum. What is interesting that these phages while possibly belonging to the Myoviridae family have bent contractile tails. A mechanism of their activity is not clear, and the authors used methods of structural cryo-EM to study tail components of this phage at nearly atomic resolution. In general, the manuscript is well written, and a lot of interesting structural information has been provided.

The authors submitted the manuscript with the title, which is confusing: they claim that the phage "requires a redox trigger for tail contraction". However, the authors did not discuss how this reaction of the oxidation will change the state of the tail elements (?), since a redox is a type of chemical reaction in which states of substrate is changed. That was neither discussed or indicated which conditions can cause the oxidation and which tail proteins will participate in this reaction and trigger the tail contraction. The title does not reflect the major topic of the manuscript that reports structural analysis of tails of the phage Milano in two states. The differences between these structures were not presented schematically to make the link to the function of this complicated protein assembly. Moreover, the authors have claimed that folds of the main tail proteins were not changed much after contraction of the tail. The second paragraph from the bottom (page 15) in the discussion is rather vague and not supported by any experiments or indications on specific conformational changes. It seems that the title must be changed to reflect the essence of this study.

It would be good to have more information on when the phage genome is injected to the host cell. The authors have written that "the flagella-tropic phages ... travel along the flagellum towards the host cell surface": when does the infection takes place? The Milano phage has apparently the same type fibres as the other flagella-tropic phages, however, it is unclear how the infection takes place and it would be good if information based on analogous phages can shed some light on this matter. Did the authors check if the Milano phages without fibres are infective? The question raised in the introduction "How such a bent and flexible tail can pierce the cell envelope?" was not considered and discussed in the conclusions in spite that the structure of the baseplate and the tail spike were obtained. How does the tail interact with the flagellar filament or with the host cell surface? Is it known which receptor is required for the phage attachment, where the receptor is it located: on the flagellum or in the host cell surface?

The conclusions in the discussion (see the second paragraph, page 14) are confusing. The authors hypothesise that "the flexible nature of the Milano tail in the extended state is an adaptation to cope with a large mechanical stress resulting from binding to a rapidly rotating flagellar filament. Such mechanical force could either trigger premature contraction and an unproductive release of the genome, or simply damage the phage particle". Possibly it should be clearer written, that the bendable tails can prevent "the premature contraction and an unproductive release of the genome" and that should be at the end of the discussion after suggesting a mechanism for the flexibility and how such flexibility can prevent malfunction of the phage. That is the essence of the manuscript, based on analysis of two conformations of these tails.

The Author Contributions section is omitted completely. Papers by Prof P. Leiman were cited a lot, but it seems that the structural analysis and interpretations were done in The Virginia Tech (Department of Biological Sciences, Blacksburg) and the University of Virginia (School of Medicine, Charlottesville).

Minor comments:

Would be good to have radial projection of the network of disulphide bonds before and after contraction and to see how the changes in the tail diameter reduce the number of the disulphide bonds in the sheath.

There is some confusion: how symmetries were defined for different elements of the base plate junction? In the introduction: the authors have written the 6-fold symmetry can be applied,

however the 3-fold symmetry was used as it was written in methods. Did the authors perform any quantitative analysis of the symmetry? Since the tail spike and base plate demonstrated C3 - symmetry how do they interact with BCP and BW1. Where are the quasi-equivalent links located?
Page 4. Last paragraph. Would be good if the authors will explain what is the CIS tube proteins.
Page 5. First paragraph. Which software was used to localise and assess disulphide bonds?
Page 5. Last paragraph. How big were changes in sizes of grooves, where the changes were observed?

Page 8. Two last paragraphs of the "Sheath protomers are locked by covalent disulphide-bonds" section. The paragraph is confusing: how much were distances between three disulphate bonds changed after contraction? It seems that in spite nearly the same conformations of the sheath protein in these two states it was not clear how big was the range of differences corresponding to ~150 Calpha atoms and where were these atoms located as indicated by higher RMSD? It is recommended that the local resolution maps should be shown as supplementary figures. Figure 2D -> it is rather difficult to see anything in this figure, the authors have to show only the front part of the helix, so the connections between protomers can be visualised. The back part of the helix should be removed.

Page 9. Second paragraph from the bottom. Why is it important to suggest the presence of the Fe ion in the structure? What is its role for the function of the spike? What does it in the pyocin?

Page 10. Where is the BW4 located? It is not shown in figures.

Page 11. The last sentence of the first paragraph on this page does not look as informative one: "The density for this connection in the pyocin cryo-EM map was also weak", since the authors did not discuss what is a role of this connection and why is it important in comparison with the pyocin.

Page 11. Figure 5. Please label in the figure 5A domains discussed in the second paragraph from the bottom.

Page 12. Second paragraph from the top. Sequence alignment for would be helpful for T4, A511, and R-type pyocin to assess what is typical and what was unusual.

Page 12. Last paragraph. What are the sizes of the STF. Many figures do not have scale bars.

Page 13. Middle paragraph. Please give the reference to a figure related to the garland.

Figure S1. A, B, and G please show only the front side of the tails, otherwise the links between subunits are not seen.

Figure S3. Please show the diffraction pattern for the noncontracted tails (yes it is bended, but for the short segment), or do not show this figure, since it is rather difficult to evaluate the differences.

Figure S4. It is confusing. The differences in A are significantly smaller compared to B. It is unclear how the authors have done the alignments of the structures: the differences apparently should be increased in opposite directions, if the central front subunits were aligned properly. Show the central section of one ring from the top. The authors have all the time the white structures above the red one in B.

Figure S5. How are these interactions reflected by the EM density maps (B)? Would be good to label the appropriate residues.

The methods: the processing should be described in more details.

Reviewer #1 (Remarks to the Author):

This work reports the first structure of a flexible, contracting tails. This is a fascinating system that departs from “classical” models in many ways. This structural study will be of great interest to the field and opens interesting research avenues that should be fruitful in coming years.

Specifically, the manuscript describes the pre- and post-conformation of the tail tube and sheath components of the tail, providing an experimental validation of a long-held hypothesis that the tail-tube is unchanged in the process. This was far from evident, especially for a tail that undergoes a transition from bent to straight. The structure also supports and adds to previous hypotheses made for Siphoviridae about the molecular basis for the tail plasticity.

In contrary, the study reveals surprising findings in the architecture of the baseplate and the presence of disulfide bonds that lock the structure in a close state and suggests a redox trigger for entry.

I very much enjoyed reading this well-written manuscript and discovering all these intriguing features. I strongly believe it will be of major interest to the field of phage research and more broadly virology.

Response: We thank the reviewer for their overall appraisal.

1.1

Here are my detailed comments to clarify some points and improve readability:

p.4 “It oligomerizes by extending the loop formed by residues 44-61 into the neighboring subunit in the hexamer.” In other phage tail tubes, the oligomerisation involves a continuous beta-barrel and the long loop contributes to inter-ring contacts. From the figures, it seems to be the case for the Milano phage too but the description doesn’t match.

Response: Yes, the hexamerization and the stacking of hexamers into the tube in Milano is pretty similar to that of other contractile tail injection systems (CIS). Besides the role in inter-ring contacts, the loop 44-61 contributes substantially to the intra-ring contacts as the interface area and binding free energy of intra-ring neighbors are increased by ~50% and ~35% respectively due to this loop. As suggested by the reviewer, the description is modified to make this fact clearer. Modified description reads “It oligomerizes by extending the loop formed by residues 44-61 into the neighboring subunit in the hexamer. The hexamers stack on top of each other with a twist of 28.21° and a rise of 34.07 Å, where the 44-61 loop contributes to the inter-ring contacts, resulting in a hollow helical tube with a ~40 Å diameter lumen.”

1.2

p.5: “However, the loop, which is absent in the Milano tube (between residues 111-112), is present in these phages, suggesting that the structural origin of grooves in these bacteriophages is different from that of Milano.”

Could this be illustrated in supplementary material? It is not clear to me that this loop is present and participating to inter-ring contacts in YSD1, 80alpha or SPP1. In YSD1 which has the most pronounced groove (Fig. S2), removal of the long loop abolishes almost all inter-ring contacts, is this the case here too?

Response: The absent loop in Milano tube (that would be between residues 111-112) fills the gap between protomers in all other CIS tubes and thus their surfaces are smooth in contrast to the Milano tube having a groovy surface. Such grooves are also present on all non-contractile phage (YSD1, 80alpha and SPP1) tubes too, but unlike Milano, they don’t lack that loop. So, the origin of grooves in

non-contractile tubes is different from Milano, i.e. different helical symmetry/organization and the presence of additional outer-domain

Unlike YSD1, the absence of this loop in Milano suggests that Milano doesn't require this loop for stabilization.

The illustration comparing the Milano tube protein with non-contractile phage (YSD1, 80alpha and SPP1) is now incorporated in supplementary material – Supp. Fig. 2H.

The grooves on the YSD1 tube are larger because its tube protein contains a larger outer domain compared to that in 80alpha and SPP1.

1.3

p.6/7: where are the 10-12 residues considered as “non-equivalent” in the superimposition located? Is this a single loop or are they scattered across the tail tube protein?

Response: The first two N-terminal and last seven C-terminal residues have not been built into the model due to the lack of clear density for them, and thus the RMSD shown is only between 127 modelled residues. The confusing word ‘equivalent’ is now replaced by ‘pruned’. So, the “non-equivalent” residues don't exist as they were never modelled.

1.4

p.8 induction of tail contraction by guanidine and DTT:

Only one small image of the straight tail is presented. More should be provided in supplementary material to allow the reader to see how straight they are in general and whether the protruding part of the tube is indeed straight without the external sheath. This is strongly suggested by the existing image but it is too small and isolated to be definitive.

Response: The broader cryo-EM micrograph and associated 2D class averages of contracted Milano tail are incorporated in Supp. Fig. 5, showing the straight morphology of the whole tail and protruding tube at the end.

1.5

p.8: “These cysteine residues are conserved in other *Agrobacterium*-infecting phages, such as 7-7-13 (Uniprotkb, J7FA80) and OLIVR4 (Uniprotkb, A0A858MST0).”

Is there a broader picture of when/where these disulfide bond locks emerged? These are a fascinating find and beg the question of how their release is mediated in vivo. Is this something specific to *Agrobacterium* or more general?

Response: We do not have any results about when/where these disulfides have emerged and how their release is mediated in nature. But here are some speculations.

The breakage of disulfides in the Milano sheath, and importantly in the baseplate, as well, during natural infection may be explained by one or more of the following plausible mechanisms. First, it is possible that the phage remains dormant until the bacteria achieve optimal growth and use all oxygen in the local environment. Disulfides are more prone for reduction in such an environment and thus the particle becomes ready to fire and infectious in such anoxic environments. Second, it is possible that the phage is waiting for the pH to drop. In that case as well, disulfides are prone to be reduced even in the presence of mild naturally occurring reducing agents and this would also unlock the sheath for contraction. Third, the presence of a specific reducing agent on the *Agrobacterium* cell surface is also

possible, which could reduce the disulfides as soon as the phage particle touches the cell-surface after translocation along flagella.

Based on the high degree of sequence similarity and conserved cysteine residues of Milano, 7-7-1 and OLIVR4 (all infecting *Agrobacterium*), the *Agrobacterium*-specific mechanism (third in aforementioned) is more likely. Another possibility is that this mechanism can be specific to the soil microenvironment near the plant root where *Agrobacterium* thrives. But we have not discussed these possibilities in the manuscript due to the lack of evidence and because of their speculative nature.

1.6

In Fig. 2, I couldn't quite follow the separation of the two "locking" cysteines and the disruption of the disulfide bond during contraction. It may be useful to change the color coding for these specific residues (Cys 47 and 182) so they can be tracked across the different panels (i.e. pre- and post-contraction).

Response: The disulfide bond, Cys47:Cys182, being broken during the contraction, is shown in yellow in Fig. 2D and by a red circle in both Fig. 2C and 2D. To make this description clearer, the disulfide bond network over the whole sheath is shown in Fig. 2E and Supp. Movie 2.

1.7

Fig. 4E: BW3 should be in the same orange shade as 4F for readability. It may be clearer if the disulfide bonds between heterotrimers are colored differently from the intra-trimer disulfide bonds. Lettering of the panel is incorrect.

Response: Changes are made in Fig. 4 as suggested.

1.8

p. 10: "As in all previously described CISs14,39,43-46, BW1 accepts the N- and C-terminal arms of the first layer of the sheath subunits, which makes the baseplate an integral part of the sheath by a sheath-like β -sheet augmentation mechanism as described above."

This is not evident in the presented figures.

Response: The said interaction is illustrated now in Supp. Fig. 6.

1.9

Given that the short-tail fiber protein displays 2 novel domains, these should be described in more details. A description of the folds and a Jones-rainbow representation would be useful. When structural comparison is mentioned (p. 12 last paragraph), it would also be important to have the key results from DALI and PDBefold searches (e.g. Z-scores, length of alignment, rmsd etc.) for the top hits even if somewhat distant.

Response: The rainbow representation for the short-tail-fiber is now shown, and the DALI search results for STF C-terminal domain (p.12 last paragraph) are now incorporated in Supp. Table 1. The description of the fold of the novel domains of STF is now added.

1.10

p.10: "The post-sheath-contraction pyocin baseplate does not contain the BCP with its helical domain (it is extruded from the baseplate by the moving tube), and the wedges of the pyocin baseplate are splayed out. Most likely, the Milano baseplate undergoes a similar conformational change upon sheath contraction."

Could the authors describe a bit more what is observed for the BP in contracted tails? Even if reconstructions have been impossible so far, there may be observations from the raw cryoEM images /

negative stain.

Does the garland need to be dissociated to give way to the tail tube?

Response: We do not have any clear evidence from cryo-EM data that provides details about the contracted baseplate. Unfortunately, the lack of evidence here limits us to only a hypothesis that the splaying out mechanism is likely for Milano. The splaying out of all six baseplate wedges has been proposed previously for Pyocin. The post-sheath-contraction baseplate of Pyocin does not contain the BCP with its helical domain (it is extruded from the baseplate by the moving tube), and the wedges of the pyocin baseplate are detached from each other and splayed out. Due to the structural and organizational similarities between Milano and Pyocin BCP, the same 'splaying out' mechanism is proposed for Milano.

The chain of garland proteins links the long tail spike (LTS) and the short tail fibers (STF), which need to splay out. Thus, we believe that the garland protein chain must unravel, and this most likely occurs in the very first step of contraction.

1.11

Discussion:

"We believe that the flexible nature of the Milano tail in the extended state is an adaptation to cope with a large mechanical stress resulting from binding to a rapidly rotating flagellar filament."

To add support to this proposal (and the following paragraph), it may be useful to mention that a similar hypothesis about mechanical stress and plasticity has been proposed for YSD1, which is also a flagellotropic phage.

Response: We agree, and now mention that this hypothesis has been proposed for YSD1.

In addition, we provide two examples:

- 1) Type 1 pili of Uropathogenic *E. coli* have a very flexible nature that can sustain high mechanical stress.
- 2) Tobacco mosaic virus has a very straight and rigid structure that is also very fragile.

Reviewer #2 (Remarks to the Author):

The paper by Sonani et al. describes an elegant structural analysis of the *Agrobacterium tumefaciens* bacteriophage Milano. The authors determined a cryo-EM reconstruction of the contractile tail of phage Milano in the bent and contracted conformations, deciphering the bent-to-straight transformation of the sheath and tube proteins. Unexpectedly, the authors found that the Milano tail sheath and baseplate are covalently linked by multiple disulfide bounds, which may enable the phage to survive the mechanical stress by binding to the host flagella. Overall, these findings broaden our understanding of covalent crosslinking in bacteriophages and shed light on the biology of flagellotropic phages that do not carry a curly fiber to wrap around the host flagellum.

Overall, I believe this paper has many strengths and advances the field of bacteriophage biology, especially for flagellotropic phages that are poorly understood. At the same time, there are several gaps that the authors should address to make this work publication quality:

Response: We thank the reviewer for the very positive overall summary.

2.1

Introduction. Can the author clarify the evidence that Milano is a flagellotropic phage like 7-7-1? Reference 2 is quite old (1977). Is Milano inferred to be a flagellotropic phage based on sequence

similarity to 7-7-1, or was this demonstrated experimentally? In general, I believe the introduction of this paper should focus more on Milano rather than generalizing Milano and 7-7-1 as if they were the same phage.

Response: The experimental evidence that Milano is a flagellotropic phage is now added as Supp. Fig. 1. The added results show that Milano could not infect an *Agrobacterium* strain devoid of the flagellar filament nor a strain having non-rotating flagella. This indicates that the rotating flagellum is a necessary requirement for Milano to infect, and thus Milano is a flagellotropic phage. The Introduction is now modified as suggested.

2.2

Page 4. End of Introduction. By all means, tone down the claim that this paper is the first experimental proof that the tail tube remains in the same conformation after contraction. Several published papers reached the same conclusion (for non-flagellotropic phages/pyocins):

- Feng Jian et. al. 2019. Cell. Cryo-EM Structure and Assembly of an Extracellular Contractile Injection System.
- Desfosses et. al. 2019. Nat Microbiol. Atomic structures of an entire contractile injection system in both the extended and contracted states.
- Peng Ge et. al. 2020. Nature. Action of a minimal contractile bactericidal nanomachine.
- Brenda Gonzalez et. al. 2021. Viruses. Structural Studies of the Phage G Tail Demonstrate an Atypical Tail Contraction.

Response: What we are talking about is the first high-resolution density map allowing a fit of the atomic model of the tail-tube in the contracted state that is almost unchanged compared to that in the extended state. The PDB/EMDB-deposited data associated with the said articles (Feng et al. 2019, Desfosses et. al. 2019, Peng et al 2020 and Brenda Gonzalez et al. 2021) along with **all other articles** published so far neither show a convincing high resolution density map, nor provide any biochemical evidence for the *assumption* that the **tail tube** (not the tail sheath subunit) retains its structure after sheath contraction. Unless a similar finding is published while this MS is in revision, our work will be the first to finally experimentally confirm this very old assumption.

2.3

Introduction and Discussion. Milano is said to be flexible and bent, but a bent tail does not imply flexibility. A boomerang is bent yet very sturdy. Can the authors clarify what biochemical/biophysical features/evidence support the claim that Milano's tail is flexible?

Response: We agree that a bent tail does not imply flexibility. The requirement to be bent-but-rigid is consistent curvature. Our observations for Milano show the opposite (see image below) – the tail displays different degrees of curvature suggesting its bent-and-flexible nature. The same can be also inferred from Fig. 1A, in which the tail of one phage particle is bent differently from the other three. This is now clarified in the paper.

2.4

Figure 4 is overwhelming. Can it be simplified to look more like Figure 3?

Response: Fig. 4 is modified as suggested.

2.5

Induction of contraction. The authors use 5 mM DTT plus 3M urea + 3 M guanidine HCl for 30 min to induce contraction. What happens if the denaturants (3M urea + 3 M guanidine HCl) are taken out of the reaction? How can they pinpoint the specificity of contraction to just DTT? Is DTT necessary and sufficient for tail contraction, or just necessary? Similarly, would a higher concentration of denaturants without DTT induce contraction?

Response: We have tested the following conditions to induce contraction: (1) 3M to 7 M Urea, (2) 3M to 7M guanidine HCl, (3) 3M Urea + 5mM DTT, (4) 3M guanidine HCl + 5 mM DTT, and (5) 5 mM DTT. Contraction can only be seen under condition# (4) 3M guanidine HCl + 5mM DTT. Neither DTT nor urea/guanidine HCl (at any of the tested concentration) alone could induce contraction. DTT is therefore necessary but not sufficient for tail-contraction. This is now clarified in the manuscript.

2.6

Did DTT and denaturants result in complete genome ejection? Did the authors observe virions with a contracted sheath and genome inside the head? Can this point be clarified and quantified accurately?

Response: The DTT and denaturant did induce the genome ejection in many phage particles but not in all particles. We have not quantified them as our focus was on studying the tail structure/morphology in the contracted state.

2.7

How were disulfide bonds identified, and what 'hard' evidence supports their existence? Were they identified based on the electron density? Can an alternative method be used to demonstrate that Milano's structural proteins are cross-linked? For instance, native gel, SDS-PAGE, or, better, mass spec analysis. I am positive the authors have seen the paper from the Zhou's lab on phage Pam3 that just came out in PANS, where MS was utilized <https://www.pnas.org/doi/10.1073/pnas.2213727120>

Response: The disulfide bonds are identified directly by visual inspection of density maps. However, at several places, the density corresponding to the disulfide bonds was not clear as the disulfide bonds are extremely sensitive to radiation damage as described in one of our earlier papers on

an unrelated system¹ (see supplementary Fig. S4 in that paper) . In the cases where the disulfide density was not clear, the C α -C α distance cutoff (<7.0 Å) was used to define the disulfide bonds². We have not checked this using Mass-spec and other methods (SDS- and Native-PAGE).

These disulfide bonds provide structural stability to Milano and we have measured and compared the stability of Milano with other phages, i.e., T4 and Chi-bacteriophages, that are both devoid of disulfide bonds. Milano particles stay intact for more than 30 sec while T4/Chi fall apart in less than 10 sec under ultrasonication-induced mechanical stress (see the Figure below which is the part of our manuscript under preparation for Milano neck and capsid structures).

Figure Description: Phage disintegration assay. NS-TEM images of disintegrated Milano, Chi and T4 bacteriophage particle by ultra-sonication. The Chi and T4 bacteriophage start disintegrating after 5-10 seconds while Milano survives more than 30 seconds of ultra-sonication. The scale bar equals 200 nm. (Procedure: Milano, Chi and T4 bacteriophages were ultrasonicated on ice by the probe ultrasonicator with pulse 1:1 sec (on:off). Samples at different time intervals (5, 10, 20 and 30 sec) were withdrawn for phage morphology analysis by NS-TEM.)

2.8

Pages 8-11. The authors provide an extensive description of the baseplate in the extended state. It would also be helpful to present the structure of the contracted baseplate. Was it observed at all? Are there significant structural differences between the two conformations?

Response: Although the structure of the contracted baseplate is beyond the scope of our manuscript, we have tried reconstructing the contracted baseplate using two different datasets but did not succeed. We speculate that this can be partly due to the non-physiological contraction conditions used. Most likely, under physiological conditions the disulfides break in a sequential manner, starting from the baseplate. The application of DTT, in contrast, likely breaks disulfides at random. In the presence of guanidine HCl the sheath likely contracts before all disulfides are reduced in the baseplate, which causes disintegration of the baseplate.

2.9

Can the authors speculate on the biology of *Agrobacterium tumefaciens* infection? When/where would the phage Milano encounter a different redox environment to trigger disulfide bond reduction/oxidation?

Response: As also stated in Response to reviewer 1's comment #1.5, we do not have any results about when/where these disulfides are released in nature. The breakage of disulfides in the Milano sheath and importantly in the baseplate, as well, during natural infection may be explained by one or more of the plausible mechanisms we suggested in response to Reviewer #1, comment #1.5.

2.10

Discussion. The authors mentioned the tape measure protein. Have they found any evidence for the tape measure protein in their reconstruction? Can this be shown/discussed?

Response: The density presumed to be from the Tape Measure Protein (TMP) in the lumen of the tail tube is quite visible in both the curved tail reconstruction and baseplate reconstruction – see the images shown below. However, the density is not good enough to fit the model and thus the Tape-measure-protein structure is not presented as a major result. Besides its density in map, the presence of TMP is also confirmed by mass-spectrometry.

2.11

Suppl Fig 7. The reconstructions obtained without (and to some extent with) a mask reveal strikingly poor correlation at the medium-to-low resolution, typically indicative of multiple conformations. This can be observed in poor(ish) electron densities, despite the FSC resolution at 0.143. Can the authors comment on this point? Can the experimental density be shown in the suppl. Info? In my experience, densities like this do not refine well against atomic models using real-space refinement (see next point)

Response: The poor FSC at medium-to-low resolution in the curved-tail (A) and baseplate reconstruction (B) can be explained by the flexible/disordered region. In the curved tail asymmetric reconstruction, the SED domain of the sheath-subunit is poorly resolved due to its different orientation at the inner and outer curves of the tail. Moreover, the tape-measure protein in the center of the tube is also not resolved because we are using overlapping tail-segments here. We believe the poor FSC results from these two regions in the map. The model for these regions has not been built.

In the baseplate reconstruction (B), the lower region of the receptor-binding complex (Residues 127-300 of STF, Residues 180-587 of LTS and garland protein), due to its long-slender shape, do not follow perfect C3 symmetry as the baseplate wedge/hub proteins do. Thus, this region is not resolved well. Secondly, the upper region of the tail connected to the baseplate starts bending and thus also does not follow C3 symmetry and is not well resolved. The poor FSC from the baseplate reconstruction is most likely due to these two regions in the map.

In the contracted sheath (C) and tube (D) reconstructions, the poor FSC can be explained by the different symmetries of the tube and sheath. Thus, the signal from the tube, having a different symmetry than that applied in the sheath reconstruction, will degrade the sheath FSC, while the signal from the sheath will degrade the FSC in the tube reconstruction.

As suggested, the density maps, model-to-map fitting and local resolution are now shown in Supp. Fig. 9.

2.12

Table 2. The authors must report the model-to-map CC, which is our best surrogate for the Rfree used in crystallography. At this resolution, all models presented in this paper and deposited in the PDB are expected to have CC>0.75. This is a vital point.

Response: The model-to-map CC values for all reconstructions are now included in Table 2.

Reviewer #3 (Remarks to the Author):

3.1

The manuscript entitled “Agrobacterium tumefaciens phage Milano requires a redox trigger for tail contraction” by Sonani and co-authors has been submitted to the Nature Communications journal. The phage Milano represents a group of flagella-tropic phages and infects hosts through binding an actively rotating flagellum. What is interesting that these phages while possibly belonging to the Myoviridae family have bent contractile tails. A mechanism of their activity is not clear, and the authors used

methods of structural cryo-EM to study tail components of this phage at nearly atomic resolution. In general, the manuscript is well written, and a lot of interesting structural information has been provided.

Response: We thank the reviewer for the positive summary.

3.2

The authors submitted the manuscript with the title, which is confusing: they claim that the phage “requires a redox trigger for tail contraction”. However, the authors did not discuss how this reaction of the oxidation will change the state of the tail elements (?), since a redox is a type of chemical reaction in which states of substrate is changed. That was neither discussed or indicated which conditions can cause the oxidation and which tail proteins will participate in this reaction and trigger the tail contraction. The title does not reflect the major topic of the manuscript that reports structural analysis of tails of the phage Milano in two states. The differences between these structures were not presented schematically to make the link to the function of this complicated protein assembly. Moreover, the authors have claimed that folds of the main tail proteins were not changed much after contraction of the tail. The second paragraph from the bottom (page 15) in the discussion is rather vague and not supported by any experiments or indications on specific conformational changes. It seems that the title must be changed to reflect the essence of this study.

Response: The title of the manuscript is now changed to “An extensive disulfide bond network prevents tail contraction in *Agrobacterium* phage Milano”. We agree with the reviewer that our results do not provide any evidence about how the disulfide bonds in the Milano structure are reduced in Nature and thus all possible mechanisms described in the discussion (second paragraph, p.15) are speculative.

We agree with the comment that the differences between the two states are not presented in some simple cartoon or schematic diagram. However, taking into account multiple previous reports stating that the sheath subunit changes its structure little upon sheath contraction, we decided to concentrate on truly novel findings – structural determinants that allow for tube bending, the conservation of the tube structure in the contracted state, the multitude of disulfides, etc. We believe that the manuscript clearly describes the structure of the two main components of the tail – the sheath and the tube – in their pre- and post-contraction states. We show that (1) The contraction of the sheath in Milano is similar to that of other CISs (Fig. 2C-2E, Supp. Videos), albeit the Milano sheath contracts to a much lesser degree (Fig. 2A). The disulfide bond network of the tail sheath is described prior and post contraction. The schematic showing disulfide bonds network in the extended and contracted sheath is now better represented in Fig. 2 and Supp. Movie 2. (2) The structure of the tube at both the subunit and polymer level is almost the same (Supp. Fig. S3, S4) between the extended and contracted states. This has only been assumed previously. Besides these two components, we explained the flexible to straight transformation of the tail based on our results. On the other hand, we do agree with the reviewer that the information about contracted baseplate, which has been unachievable so far, is not provided.

3.3

It would be good to have more information on when the phage genome is injected to the host cell. The authors have written that “the flagella-tropic phages ... travel along the flagellum towards the host cell surface”: when does the infection takes place? The Milano phage has apparently the same type fibres as the other flagella-tropic phages, however, it is unclear how the infection takes place and it would be good if information based on analogous phages can shed some light on this matter. Did the authors check if the Milano phages without fibres are infective?

Response: Some of us have proposed the steps in the infection cycle of *Agrobacterium* phage 7-7-1 (the closest homolog of Milano) in a previous article³ - “The phage initially attaches to the flagellar filament and is propelled down toward the cell surface by clockwise flagellar rotation. The phage then attaches to secondary cell surface receptors - the lipopolysaccharides, establishes contact with the cell surface and ejects its DNA into the cell by tail contraction”. From the high-resolution structures described in the present manuscript, we now know that the disulfides in both the tail-sheath and baseplate need to be broken to induce contraction and subsequent genome-release. However, we do not know whether Milano also uses lipopolysaccharides as secondary receptors.

The tail-fiber network of Milano (and its homolog 7-7-1) is clearly different from other flagellotropic phages as it contains six short tail-fibers compared to the single very long tail fiber in flagellotropic siphophages YSD1 and Chi.

We have not checked if Milano particles devoid of tail fibers are infectious.

3.4

The question raised in the introduction “How such a bent and flexible tail can pierce the cell envelope?” was not considered and discussed in the conclusions in spite that the structure of the baseplate and the tail spike were obtained. How does the tail interact with the flagellar filament or with the host cell surface?

Response: The flexible-to-rigid transformation of the tail is the answer to this question. The following text in the revised manuscript discusses the answer to this question.

‘Grooves on the tube and electrostatic potentials of tube/sheath surfaces provide the Milano tail with a dual physical nature. The Milano tail is flexible enough (like non-contractile tails) to resist strong mechanical forces in the extended state, and rigid in the contracted state, allowing it to pierce the cell membrane.’

Our results do not show any interactions of tail fibers or tail spike with the host cell and flagella.

3.5

Is it known which receptor is required for the phage attachment, where the receptor is it located: on the flagellum or in the host cell surface?

Response: The inability of Milano to infect strains of *Agrobacterium tumefaciens* that lack flagella or that are non-motile (new Supp. Fig. 1) indicates that the primary receptor must be on the flagellar filament. Based on our analysis of *Agrobacterium* phage 7-7-1 (closest homolog of Milano), the secondary receptors, lipopolysaccharides on the cell surface, are also crucial for the infection³. However, the secondary receptor for Milano, which must be located closer to the cell surface, has not been identified.

3.6

The conclusions in the discussion (see the second paragraph, page 14) are confusing. The authors hypothesise that “the flexible nature of the Milano tail in the extended state is an adaptation to cope with a large mechanical stress resulting from binding to a rapidly rotating flagellar filament. Such mechanical force could either trigger premature contraction and an unproductive release of the genome, or simply damage the phage particle”. Possibly it should be clearer written, that the bendable tails can prevent “the premature contraction and an unproductive release of the genome” and that

should be at the end of the discussion after suggesting a mechanism for the flexibility and how such flexibility can prevent malfunction of the phage. That is the essence of the manuscript, based on analysis of two conformations of these tails.

Response: The discussion has been modified as suggested. The specific sentence now reads: 'A flexible and bendable tail can prevent the premature trigger of contraction, an unproductive release of the genome, or damage to the phage particle, that could otherwise happen under the mechanical stress.' The discussion about 'how such flexible tail can prevent the structural damage and malfunction' is now moved to the end of discussion of tail structure.

3.7

The Author Contributions section is omitted completely. Papers by Prof P. Leiman were cited a lot, but it seems that the structural analysis and interpretations were done in The Virginia Tech (Department of Biological Sciences, Blacksburg) and the University of Virginia (School of Medicine, Charlottesville).

Response: We have now added the author's contribution section in the revised manuscript. Which reads "E.H.E., B.E.S and R.R.S. designed the study, while P.G.L. provided additional suggestions for experiments. The phage sample preparation and plaque assay were conducted by N.C.E., A.A.H., A.L.S. and R.J.K. under the guidance of B.E.S. Cryo-EM grid preparation, experiment, image-analysis, reconstructions, and model building were conducted by R.R.S., with suggestions from E.H.E., P.G.L., F.W. and M.A.B.K. The phage contraction experiment was performed by R.R.S with suggestions from P.G.L. and E.H.E. Manuscript was prepared by R.R.S, E.H.E, P.G.L and B.E.S. Additional suggestions on the manuscript were provided by F.W. and M.A.B.K."

3.8

Minor comments:

Would be good to have radial projection of the network of disulphide bonds before and after contraction and to see how the changes in the tail diameter reduce the number of the disulphide bonds in the sheath.

Response: The radial representation and projection of disulfides bonds are included in Fig. 2 and Supp. Video 2.

3.9

There is some confusion: how symmetries were defined for different elements of the base plate junction? In the introduction: the authors have written the 6-fold symmetry can be applied, however the 3-fold symmetry was used as it was written in methods. Did the authors perform any quantitative analysis of the symmetry? Since the tail spike and base plate demonstrated C3 -symmetry how do they interact with BCP and BW1. Where are the quasi-equivalent links located?

Response: The C6 symmetry mentioned in the introduction is for tail-tube and tail-sheath. But, as the reviewer points out, the baseplate centerpiece (BCP), baseplate wedge region (BW) (containing baseplate wedge proteins 1, 2 and 3) and receptor binding complex (RBC) also possess C6 symmetry. In contrast, the baseplate hub (BH) and central spike (BCS) only have C3 symmetry. We initially imposed C6 symmetry for the whole baseplate reconstruction, resulting in a map having well-resolved density for

three components (BCP, BW, RBC) and poorly resolved density for BH and CS. Lowering the symmetry to C3 gave reasonably good density for BH and CS, too and allowed de novo model building. Thus, for the whole baseplate structure, the highest symmetry that can be applied is C3. No quantitative analysis for the symmetry has been performed.

Similar to T4, the “tandem tube domain” of BH serves as the symmetry adaptor (the quasi-equivalent link). This mechanism is first described here: Kanamaru, S. et al. Structure of the cell-puncturing device of bacteriophage T4. *Nature* **415**, 553-557 (2002)⁴.

3.10

Page 4. Last paragraph. Would be good if the authors will explain what is the CIS tube proteins.

Response: The abbreviation CIS is now defined at the first instance in the manuscript.

3.11

Page 5. First paragraph. Which software was used to localise and assess disulphide bonds?

Response: The disulfide bonds were identified manually based on high-resolution density maps. However, in several places, the density corresponding to the disulfide bonds is not clear as the disulfides are very prone to the radiation damage as described in a paper on an unrelated system¹ (see supplementary Fig. S4 in that paper). Where the disulfide density is not clear, the C α -C α distance cutoff of 7.0 Å is used for defining the disulfide bonds². (See Response to comment #2.7 of Reviewer #2 for more details).

3.12

Page 5. Last paragraph. How big were changes in sizes of grooves, where the changes were observed?

Response: We assume that the reviewer here is asking about the difference in groove sizes between inner and outer curves of the bent Milano tail. As shown in Fig. 2G, grooves on the outer curve are substantially bigger than those on the inner curve of the tube. The distance between subunits in upper and lower ring (which is lining the groove) is increased ~ 1 Å, for instance the distance between C α s of Pro33 of the lower ring subunit and Thr51 of the upper ring subunit is increased from ~ 5.7 Å at the inner curve to ~ 6.6 Å at the outer curve.

3.13a

Page 8. Two last paragraphs of the “Sheath protomers are locked by covalent disulphide-bonds” section. The paragraph is confusing: how much were distances between three disulphate bonds changed after contraction?

Response: As discussed in our Response to Reviewer #3’s comment #3.11, we have considered the Cys C α -C α distances for disulfide bond cut-off. The C α -C α distances for three Cys-Cys pairs in the extended and contracted state are shown below. In the first two bonds, the C α -C α distance is increased

by ~ 1.0 and ~ 1.5 Å, respectively, whereas the increase in the third one is negligible (~ 0.2 Å). A C α -C α distance greater than 7.0 Å is considered a broken disulfide bond.

		C α -C α distance (Å)	
		Extended	Contracted
1	Cys14:Cys244	4.7	5.7
2	Cys47:Cys182	6.1	7.5
3	Cys216:Cys327	5.3	5.4

13b

It seems that in spite nearly the same conformations of the sheath protein in these two states it was not clear how big was the range of differences corresponding to ~ 150 C α atoms and where were these atoms located as indicated by higher RMSD?

Response: As shown in the following Figure, the larger RMSD is mainly caused by the N- and C-terminal strands (~ 50 residues) invading into the neighboring subunit's HSD, and less by the HSD itself (~ 100 residues).

3.14

It is recommended that the local resolution maps should be shown as supplementary figures.

Response: Local resolution maps are now shown in Supp. Fig. 9.

3.15

Figure 2D -> it is rather difficult to see anything in this figure, the authors have to show only the front part of the helix, so the connections between protomers can be visualised. The back part of the helix should be removed.

Response: Fig. 2D has been modified as suggested.

3.16

Page 9. Second paragraph from the bottom. Why is it important to suggest the presence of the Fe ion in the structure? What is its role for the function of the spike? What does it in the pyocin?

Response: The presence of an Fe ion in the central spike of phage P2, phi92⁵ and Mu⁶ (and now in Milano) have been proposed to provide structural stability to the central spike oligomer which is required for physical piercing of the cell envelope. A sentence proposing the functional relevance of the Fe ion has now been added to the manuscript.

3.17

Page 10. Where is the BW4 located? It is not shown in figures.

Response: BW4 is present in T4 and pyocin but not in the Milano baseplate. The C-terminal domain of Milano BW3 is equivalent to BW4 of T4/pyocin.

3.18

Page 11. The last sentence of the first paragraph on this page does not look as informative one: "The density for this connection in the pyocin cryo-EM map was also weak", since the authors did not discuss what is a role of this connection and why is it important in comparison with the pyocin.

Response: That sentence is now removed.

3.19

Page 11. Figure 5. Please label in the figure 5A domains discussed in the second paragraph from the bottom.

Response: The labels for LTS domains in Fig. 5A have been added.

3.20

Page 12. Second paragraph from the top. Sequence alignment for would be helpful for T4, A511, and R-type pyocin to assess what is typical and what was unusual.

Response: We are not sure if we understand this comment correctly. The second paragraph on Page 12 is strictly a structural description of how the Tail Spike (TSP) is attached to the baseplate and how it is different from what is seen in T4, A511 and R-type pyocin. In our opinion, the sequence

alignment in such a system will not help, especially when the homology among these proteins is only apparent at the structural level.

3.21

Page 12. Last paragraph. What are the sizes of the STF. Many figures do not have scale bars.

Response: The short tail fiber (STF) is an ~170 Å long slender fiber-like structure. It is a trimer of the 300 residue gp31 proteins, each having a molecular weight of ~32 kDa. Scalebars in Fig. 2A, Fig. 5 and several Supp. figures have been added.

3.22

Page 13. Middle paragraph. Please give the reference to a figure related to the garland.

Response: Relevant figures are cited now.

3.22

Figure S1. A, B, and G please show only the front side of the tails, otherwise the links between subunits are not seen.

Response: All tail/tube images in Supp. Fig. 1 are now shown with only the front side of the tail, as suggested.

3.23

Figure S3. Please show the diffraction pattern for the noncontracted tails (yes it is bended, but for the short segment), or do not show this figure, since it is rather difficult to evaluate the differences.

Response: Averaged power spectrum of extended bent tail segments has now been added.

3.24

Figure S4. It is confusing. The differences in A are significantly smaller compared to B. It is unclear how the authors have done the alignments of the structures: the differences apparently should be increased in opposite directions, if the central front subunits were aligned properly. Show the central section of one ring from the top. The authors have all the time the white structures above the red one in B.

Response: The tube structures are now superimposed using the central ring as suggested. The updated RMSD values for tube subunit and tube-oligomer (stack of 3 hexameric rings) is now ~0.7 Å and ~1.6 Å, respectively. The relevant text has been modified.

3.25

Figure S5. How are these interactions reflected by the EM density maps (B)? Would be good to label the appropriate residues.

Response: Density map and residue labels have been added to this Supp. Figure.

3.26

The methods: the processing should be described in more details.

Response: More details about cryo-EM data processing have been added in Methods.

References

1. Pieri, L. et al. Atomic structure of Lanreotide nanotubes revealed by cryo-EM. *Proceedings of the National Academy of Sciences* **119**, e2120346119 (2022).
2. Gao, X., Dong, X., Li, X., Liu, Z. & Liu, H. Prediction of disulfide bond engineering sites using a machine learning method. *Scientific Reports* **10**, 1-9 (2020).
3. Gonzalez, F., Helm, R.F., Broadway, K.M. & Scharf, B.E. More than rotating flagella: Lipopolysaccharide as a secondary receptor for flagellotropic phage 7-7-1. *Journal of bacteriology* **200**, e00363-18 (2018).
4. Kanamaru, S. et al. Structure of the cell-puncturing device of bacteriophage T4. *Nature* **415**, 553-557 (2002).
5. Browning, C., Shneider, M.M., Bowman, V.D., Schwarzer, D. & Leiman, P.G. Phage pierces the host cell membrane with the iron-loaded spike. *Structure* **20**, 326-39 (2012).
6. Harada, K. et al. Crystal structure of the C-terminal domain of Mu phage central spike and functions of bound calcium ion. *Biochim Biophys Acta* **1834**, 284-91 (2013).

REVIEWER COMMENTS

Reviewer #1 (Remarks to the Author):

The authors answered some of my questions and comments in the revised manuscript. A few key points still need to be addressed:

While the authors have illustrated the differences between the tail tube proteins of the Milano phage and non-contractile tails with an additional figure panel, it remains unclear what the basis for flexibility is. Is this simply a matter of a reduced buried surface or a different nature of anchor points? In Siphophage-like tails, the rings are held by one or two isolated anchor points per subunit that involve the long loop and/or C-terminus (e.g. Fig. 3 in doi: 10.1042/BST20210799). Does the inter-ring disulfide bond have the same role as the C-terminus in these other flexible tails? An analysis of buried surface areas with PISA may help clarify this point and/or a figure where loop 44-61 and the N-terminus containing Cys3 are removed (i.e. are the rings still connected?).

"It oligomerizes by extending the loop formed by residues 44-61 into the neighboring subunit in the hexamer."

This is still either incomplete or incorrect. Please mention that the rings are primarily formed by a continuous 24-strand beta-barrel before describing how the extended loop also contributes to the oligomerisation.

The exact position of Cys47 and Cys182 before and after contraction remains unclear in the updated figure 2. From the overall views in Fig. 2 C & D, it is impossible to follow how the two residues moved apart. From the previous version of the manuscript, I pictured a significant separation between the two residues upon contraction. However, the disulphide bond looks intact in Fig.2D and movie 2. Given the relatively small movements, a zoomed view of these two residues in the pre- and post-contraction structures should be provided with the electron density map (illustrating the movie description "inter-Ca distances between Cysteines is increased to ~7.5 Å and their 'SG' atoms are separated by the density blob corresponding to the DTT")

Rebuttal: "We do not have any results about when/where these disulfides have emerged and how their release is mediated in nature."

To be more precise, the suggestion was to generate sequence alignments to follow the conservation of the key cysteines proposed to be functionally important. It should then be possible to comment on their presence/absence in different phylogenetic clades. If not possible for lack of sequence similarity, please mention it in the text where conservation of cysteines is already described (p.8).

Reviewer #2 (Remarks to the Author):

The authors have amended the original paper in response to the reviewer's comments. Sup Fig 1 is very informative, providing valuable biological context to the paper. This figure could easily be part of the main text. However, the rest of the paper is only moderately improved, and significant points raised by the reviewers remain unanswered.

First and foremost, there continues to be no hard evidence that the phage Milano tail is crosslinked, as the authors describe in this paper. This is a major point. Suppose the biology of phage Milano infection and/or Agrobacterium was better understood, and evidence that this phage uses redox potential to oxidize/reduce cys residues during infection was in the literature. In that

case, the proposed structural model could be sufficient to prove the point. However, without a priori knowledge, if the authors use structural biology to demonstrate the existence of disulfide bonds and claim that some of these bonds must be reduced for contraction to occur, then the bar is higher. The authors cannot just assume the tail is crosslinked without providing reproducible and convincing structural and biochemical evidence. It is a matter of scientific rigor.

I continue to see no hard evidence in this paper that the disulfide bonds are real and form/break during infection. The authors should have responded better to the reviewers' comments instead of talking them out. First, the authors did not provide MS evidence as requested by the reviewer and shown for Pam3 in a recent PNAS paper (<https://www.pnas.org/doi/10.1073/pnas.2213727120>). Second, they failed to show representative electron densities claiming the density is too weak (three reviewers asked for this). Third, they admitted that reducing agent is necessary but not sufficient to break disulfide bridges. They have to denature the tail with 3M denaturant in the presence of high DTT to reduce the alleged covalent bonds. There is no such concentration of denaturant in nature (unless dealing with a thermophilic organism), arguing against the biological significance of this observation. A movie that animates the alleged disulfide bonds is by no means structural evidence for such a finding.

Overall, I do not think the authors have made a reasonable effort to address this crucial point that all reviewers, with lesser or greater emphasis, have pointed out.

A second point emerged from reading the paper and the response to the reviewers' comments. Overall, I believe that the paper would benefit from a greater focus on the contractile sheath-rigid tube machinery rather than the baseplate components. It is not clear if the C6 symmetry is genuine and/or why C3 is used. The authors should provide more detailed figures on the conformational changes of the sheath. Additionally, there is a notable absence of figures depicting the contracted neck-collar and sheath interfaces in the study. While the authors examined the Milano phage's contractile tail, their analysis did not cover the entire process from neck to sheath and sheath to baseplate.

Minor points

Figure 1 (D), Cys82, and the line pointing to the residue are miss-aligned. Why are res 111-112 marked with a blue sphere? Similarly, what does Figure 1(H) convey? There is no explanation in the main text. It would be helpful to report the RMSD and major differences between the tube subunits.

Figure 2 (A), please remove the red dots or explain what they represent. The yellow circle in Figure 2 (B) has to be explained in the legend. Figures 1 and 2 could be combined to show the most critical findings, and it may be possible to eliminate some redundancy in Figures 2 (C), (D), and (E).

Figures 2 (C) and (D). It is unclear what the red spheres and circles represent.

Figure 2 (F), only the right panel is necessary.

Figure 4. There is no description of the red sphere in the legend. An explanation for the reader would be helpful.

Figure 5 (B). It is unclear what the red and blue squares represent.

Reviewer #3 (Remarks to the Author):

The authors have addressed nearly all questions and comments raised by the reviewers. Possibly, some expressions could be slightly modified linguistically, but they will not change the essence of the submitted manuscript. The methods on EM were extended and necessary information was provided, the figures were improved. I believe the manuscript is suitable for the publication.

RESPONSE TO REVIEWERS

Reviewer #1 (Remarks to the Author):

Comment: The authors answered some of my questions and comments in the revised manuscript. A few key points still need to be addressed:

While the authors have illustrated the differences between the tail tube proteins of the Milano phage and non-contractile tails with an additional figure panel, it remains unclear what the basis for flexibility is. Is this simply a matter of a reduced buried surface or a different nature of anchor points? In Siphophage-like tails, the rings are held by one or two isolated anchor points per subunit that involve the long loop and/or C-terminus (e.g. Fig. 3 in doi: 10.1042/BST20210799).

Does the inter-ring disulfide bond have the same role as the C-terminus in these other flexible tails? An analysis of buried surface areas with PISA may help clarify this point and/or a figure where loop 44-61 and the N-terminus containing Cys3 are removed (i.e. are the rings still connected?).

Response: The flexibility in Milano can be explained by the significantly reduced interfacial (buried surface) area ($\sim 8,500 \text{ \AA}^2$) between two hexameric rings compared to other contractile tail systems (CIS) tubes, i.e. T4 ($\sim 16,600 \text{ \AA}^2$), T6SS ($\sim 14,600 \text{ \AA}^2$), RTP ($\sim 15,100 \text{ \AA}^2$), PVC ($\sim 16,100 \text{ \AA}^2$) and AFP ($\sim 15,900 \text{ \AA}^2$), as calculated by the PISA server¹. An important question is why the interfacial area in the Milano tube is so much smaller compared to tubes of other CIS. The answer lies in Suppl. Fig. 2C of the revised manuscript. The Milano tube protein lacks a specific loop, which is present in other CIS tubes. This loop (in bacteriophage T4, this loop is residues 141-152; in T6SS, it is residues 127-140) is an important element holding hexameric rings together, in addition to the three anchor points highlighted in Fig. 3 of doi: 10.1042/BST20210799. The absence of this loop in Milano results in a greatly reduced interface and in grooves (Shown in Fig. 1E right panel by red arrows), which allows for flexibility. We directly observe that these grooves are getting smaller and larger at the inner and outer surface of the curved tube, respectively (Suppl. Fig. 2E), showing that these grooves enable the tube bending.

The inter-ring disulfide bond in the Milano tube seems to mimic the C-terminal extension that invades the upper hexamer in the tubes of Siphophage, 80 α and SPP1 to strengthen the inter-ring association. The presence of this disulfide bond increases the free energy of the inter-ring complex from 32.4 to 52.1 kCal/mol as calculated by PISA¹.

As suggested by the reviewer, we now include:

1. The point about reduced interface area in Milano with surface area calculated using PISA highlighting the role of N-terminal residues 1-7 and the loop 44-61, with an additional figure (Supp. Fig. 1B).
2. A brief discussion of the structural importance of the inter-ring disulfide bond.

Comment: "It oligomerizes by extending the loop formed by residues 44-61 into the neighboring subunit in the hexamer." This is still either incomplete or incorrect. Please mention that the rings are primarily formed by a continuous 24-strand beta-barrel before describing how the extended loop also contributes to the oligomerisation.

Response: The description has been corrected as per the suggestion.

Comment: The exact position of Cys47 and Cys182 before and after contraction remains unclear in the updated figure 2. From the overall views in Fig. 2 C & D, it is impossible to follow how the two residues moved apart. From the previous version of the manuscript, I pictured a significant separation between the two residues upon contraction. However, the disulphide bond looks intact in Fig.2D and movie 2. Given the relatively small movements, a zoomed view of these two residues in the pre- and post-contraction structures should be provided with the electron density map (illustrating the movie description “inter-C α distances between Cysteines is increased to ~ 7.5 Å and their ‘SG’ atoms are separated by the density blob corresponding to the DTT”)

Response: The suggested inclusion of a zoomed view of the Cys47-Cys182 bond with density maps in pre- and post-contracted states of tail-sheath is now included in Figure 2.

Comment: Rebuttal: “We do not have any results about when/where these disulfides have emerged and how their release is mediated in nature.”

To be more precise, the suggestion was to generate sequence alignments to follow the conservation of the key cysteines proposed to be functionally important. It should then be possible to comment on their presence/absence in different phylogenetic clades. If not possible for lack of sequence similarity, please mention it in the text where conservation of cysteines is already described (p.8).

Response: As mentioned in our previous response, the sheath protein sequence similarities among phages infecting different bacteria are minimal. Due to this fact, sequence-based alignment has not been possible. We therefore have generated structure-based alignments using DALI and checked the presence/absence of the cysteines at structurally equivalent position in sheath proteins of different bacteriophages (Suppl. Fig. 7). The presence of these cysteines is associated with the bacteriophage clade infecting *Agrobacterium*. This fact is now discussed in the manuscript.

Reviewer #2 (Remarks to the Author):

Comment: The authors have amended the original paper in response to the reviewer’s comments. Sup Fig 1 is very informative, providing valuable biological context to the paper. This figure could easily be part of the main text. However, the rest of the paper is only moderately improved, and significant points raised by the reviewers remain unanswered. First and foremost, there continues to be no hard evidence that the phage Milano tail is crosslinked, as the authors describe in this paper. This is a major point. Suppose the biology of phage Milano infection and/or *Agrobacterium* was better understood, and evidence that this phage uses redox potential to oxidize/reduce cys residues during infection was in the literature. In that case, the proposed structural model could be sufficient to prove the point. However, without a priori knowledge, if the authors use structural biology to demonstrate the existence of disulfide bonds and claim that some of these bonds must be reduced for contraction to occur, then the bar is higher. The authors cannot just assume the tail is crosslinked without providing reproducible and convincing structural and biochemical evidence. It is a matter of scientific rigor.

Response: Part of Suppl. Fig 1 is now shifted to the main Fig. 1. The “hard” evidence (mass spectrometry data - Suppl. Fig. 6; Suppl. Table 1) for the disulfide bonds is now provided in the revised manuscript. We believe that, in addition to this mass spectrometry data, two other lines of evidence support our claim

that the Milano tail is a disulfide-crosslinked assembly and that these bonds need to be released in order to induce the tail-contraction:

1. Biochemical evidence shows the necessity of DTT (a reducing agent) to induce the contraction in Milano tail.
2. The overall structure of Milano is robust and can survive much more mechanical stress as compared to the other bacteriophages, and this is due to these disulfide bonds. But in the presence of DTT (which released all disulfide bonds), Milano becomes much more susceptible to mechanical stress. This is described in detail in the Supplementary Fig. 6 of our recent paper:

Sonani, R. R., Esteves, N. C., Horton, A. A., Kelly, R. J., Sebastian, A. L., Wang, F., ... & Egelman, E. H. (2023). Neck and capsid architecture of the robust *Agrobacterium* phage Milano. *Communications Biology*, 6(1), 921.

Comment: I continue to see no hard evidence in this paper that the disulfide bonds are real and form/break during infection. The authors should have responded better to the reviewers' comments instead of talking them out. First, the authors did not provide MS evidence as requested by the reviewer and shown for Pam3 in a recent PNAS paper (<https://www.pnas.org/doi/10.1073/pnas.2213727120>).

Response: The hard evidence for the disulfide bonds (disulfide mapping by mass spectrometry - Suppl. Fig. 6; Suppl. Table 1) is now provided in the revised manuscript.

Comment: Second, they failed to show representative electron densities claiming the density is too weak (three reviewers asked for this).

Response: The densities for disulfide bonds, including ones with weaker density, are now shown in Supp. Fig. 3. It should be noted that disulfide bonds are remarkably sensitive to radiation, either x-ray or electrons. We have previously shown at 2.5 Å resolution that the sensitivity of the disulfide bond to radiation is dependent on the local environment, so that with the same radiation dose the density for some disulfide bonds is almost totally lost while others are retained².

Comment: Third, they admitted that reducing agent is necessary but not sufficient to break disulfide bridges. They have to denature the tail with 3M denaturant in the presence of high DTT to reduce the alleged covalent bonds. There is no such concentration of denaturant in nature (unless dealing with a thermophilic organism), arguing against the biological significance of this observation. A movie that animates the alleged disulfide bonds is by no means structural evidence for such a finding.

Response: Yes, DTT alone is not sufficient to induce tail-contraction but is required in addition to chaotropic agents (GnHCl or Urea). These chaotropic agents are widely used to induce contraction *in vitro* in many Myophages³⁻⁶, regardless of the mesophilic/thermophilic host. The key point is that such chaotropic agents alone cannot induce contraction in Milano. We think that the chaotropic agent induces conformational changes in the baseplate triggering contraction, which, in nature, would be induced upon the contact of the phage particle with the host surface.

Comment: Overall, I do not think the authors have made a reasonable effort to address this crucial point that all reviewers, with lesser or greater emphasis, have pointed out.

Response: We agree and believe that this is now addressed in the revised paper with mass spectrometry results confirming the predicted disulfide bonds and cryo-EM maps directly showing the density from these disulfide bonds.

Comment: A second point emerged from reading the paper and the response to the reviewers' comments. Overall, I believe that the paper would benefit from a greater focus on the contractile sheath-rigid tube machinery rather than the baseplate components.

Response: We believe that the structural and organizational description of baseplate components along with the tail makes sense for the following reasons:

1. *Continuous network of disulfide bonds:* The disulfide bond network in the Milano tail continues through the baseplate and receptor binding complex and between them. This makes the Milano particle resistant to "normal" contraction induced by chaotropic agents. The disulfide bonds in both the baseplate and tail-sheath must be reduced to induce tail-contraction. In order to discuss the contraction, it is thus necessary to also describe disulphide bonds in the baseplate.
2. *Direct attachment of Tail-Spike to the Tail-sheath:* Generally, Tail Spikes and the receptor binding proteins (RBPs) are bound to the baseplate. Surprisingly, the Tail Spikes in Milano are directly bound to the Tail-Sheath at several points, which has been another motivation to include the baseplate and RBPs structure along with the tail.
3. *The presence of a unique baseplate central piece protein:* The unique baseplate center piece protein in Milano crosslinks the distal end of the tail to the baseplate wedge, motivation for us to describe the baseplate wedge structure.

Thus, the tail and baseplate are connected unusually and uniquely in Milano and their structural description together is important.

Comment: It is not clear if the C6 symmetry is genuine and/or why C3 is used.

Response: Reviewer 3 has asked a similar question in the previous review, which is pasted below along with our response.

Previous comment # 3.9: There is some confusion: how symmetries were defined for different elements of the base plate junction? In the introduction: the authors have written the 6-fold symmetry can be applied, however the 3-fold symmetry was used as it was written in methods. Did the authors perform any quantitative analysis of the symmetry? Since the tail spike and base plate demonstrated C3 - symmetry how do they interact with BCP and BW1. Where are the quasi-equivalent links located?

Our previous response: The C6 symmetry mentioned in the introduction is for tail-tube and tail-sheath. But, as the reviewer points out, the baseplate centerpiece (BCP), baseplate wedge region (BW) (containing baseplate wedge proteins 1, 2 and 3) and receptor binding complex (RBC) also possess C6 symmetry. In contrast, the baseplate hub (BH) and central spike (BCS) only have C3 symmetry. We initially imposed C6 symmetry for the whole baseplate reconstruction, resulting in a map having well-resolved density for three components (BCP, BW, RBC) and poorly resolved density for BH and CS. Lowering the symmetry to C3 gave reasonably good density for BH and CS, too and allowed de novo model building. Thus, for the whole baseplate structure, the highest symmetry that can be applied is C3. No quantitative analysis for the symmetry has been performed. Similar to T4, the "tandem tube domain"

*of BH serves as the symmetry adaptor (the quasi-equivalent link). This mechanism is first described here: Kanamaru, S. et al. Structure of the cell-puncturing device of bacteriophage T4. Nature 415, 553-557 (2002)*⁷.

As explained, the C3 symmetry was applied based on our observation that the C6-symmetrization has blurred out the density corresponding to BH and CS, whereas reducing the symmetry to C3 resolves this density reasonably well and allowed *de novo* model building in this region of density. Moreover, the BH and CS of other myophages like T4, XM1 and A511 are also characterized to adopt C3 symmetry suggesting it as consensus symmetry for BH and CS.

Comment: The authors should provide more detailed figures on the conformational changes of the sheath.

Response: The conformational change in the Tail-Sheath is now shown in more detail in Figure 2. Figure 2 now clearly shows:

1. Location of disulfide bonds in both states,
2. Close-up view of the disulfide C47:C182 with density in both normal and contracted states
3. Changes in sheath protomer orientation with respect to the helical axis.

The modified Figure 2 along with the Supplementary Movie (showing morphed transition from normal to contracted state) now describes the structural transition in more detail.

Comment: Additionally, there is a notable absence of figures depicting the contracted neck-collar and sheath interfaces in the study. While the authors examined the Milano phage's contractile tail, their analysis did not cover the entire process from neck to sheath and sheath to baseplate.

Response: We believe that the structure of neck-collar/sheath junction (in pre-contracted state) would add neither to “tail flexibility/rigidity” nor to “phenomenon of contraction” – which are two central aspects of the paper and these should be described separately. We describe the structure of the tail-neck junction in our recently published article: *Sonani, R. R., Esteves, N. C., Horton, A. A., Kelly, R. J., Sebastian, A. L., Wang, F., ... & Egelman, E. H. (2023). Neck and capsid architecture of the robust Agrobacterium phage Milano. Communications Biology, 6(1), 921.*

More importantly, the structure of neck-collar/sheath junction (in post-contracted state) is unachievable by our current experimental approach using chemical-induced contraction. As with our pre-contracted collar structure, the neck-collar/sheath interaction again involves a number of disulfide bonds. Since DTT would reduce all disulfide bonds at the same time, the contracted collar structure would be disrupted unnaturally in chemically-contracted particles. Due to this problem, we are not sure if the structure of other regions than the tail-tube and tail-sheath (e.g., collar and baseplate) of Milano in contracted state are solvable by our current approach.

Minor points

Comment: Figure 1 (D), Cys82, and the line pointing to the residue are miss-aligned.

Response: It is corrected now.

Comment: Why are res 111-112 marked with a blue sphere?

Response: It is described in the relevant figure legend. *“Cyan ball indicates the site (residues 111-112) where a loop present in other CIS is missing in Milano gp21.”*

Comment: Similarly, what does Figure 1(H) convey? There is no explanation in the main text. It would be helpful to report the RMSD and major differences between the tube subunits.

Response: Fig 1H is now removed and the RMSD is now added to the text.

Comment: Figure 2 (A), please remove the red dots or explain what they represent.

Response: Red dots are now removed from Fig. 2A.

Comment: The yellow circle in Figure 2 (B) has to be explained in the legend.

Response: Figure 2B is removed as part of major changes in Fig 1 and 2 as suggested by reviewers.

Comment: Figures 1 and 2 could be combined to show the most critical findings, and it may be possible to eliminate some redundancy in Figures 2 (C), (D), and (E).

Response: Figure 1, 2 and Suppl. Fig. 1 have been modified substantially for better presentation of important points and removal of redundancy in response to comments of Reviewer 1 and Reviewer 2.

Comment: Figures 2 (C) and (D). It is unclear what the red spheres and circles represent.

Response: Figure 2B is removed as part of major changes in Fig 1 and 2.

Comment: Figure 2 (F), only the right panel is necessary.

Response: It is modified now as suggested.

Comment: Figure 4. There is no description of the red sphere in the legend. An explanation for the reader would be helpful.

Response: The red and yellow spheres in Figure 4 are now described in the figure legend.

Figure 5 (B). It is unclear what the red and blue squares represent.

Response: Red and blue squares are now described in the figure legend.

Reviewer #3 (Remarks to the Author):

The authors have addressed nearly all questions and comments raised by the reviewers. Possibly, some expressions could be slightly modified linguistically, but they will not change the essence of the submitted manuscript. The methods on EM were extended and necessary information was provided, the figures were improved. I believe the manuscript is suitable for the publication.

Response: Thank you for the overall appraisal and your critical comments that helped us improve the manuscript.

References

1. Krissinel, E. & Henrick, K. Inference of macromolecular assemblies from crystalline state. *Journal of molecular biology* **372**, 774-797 (2007).
2. Pieri, L. et al. Atomic structure of Lanreotide nanotubes revealed by cryo-EM. *Proc Natl Acad Sci U S A* **119**(2022).
3. Leiman, P.G., Chipman, P.R., Kostyuchenko, V.A., Mesyanzhinov, V.V. & Rossmann, M.G. Three-dimensional rearrangement of proteins in the tail of bacteriophage T4 on infection of its host. *Cell* **118**, 419-29 (2004).
4. Arisaka, F., Engel, J. & Klump, H. Contraction and dissociation of the bacteriophage T4 tail sheath induced by heat and urea. *Prog Clin Biol Res* **64**, 365-79 (1981).
5. Guerrero-Ferreira, R.C. et al. Structure and transformation of bacteriophage A511 baseplate and tail upon infection of *Listeria* cells. *The EMBO Journal* **38**, e99455 (2019).
6. To, C.M., Kellenberger, E. & Eisenstark, A. Disassembly of T-even bacteriophage into structural parts and subunits. *J Mol Biol* **46**, 493-511 (1969).
7. Kanamaru, S. et al. Structure of the cell-puncturing device of bacteriophage T4. *Nature* **415**, 553-557 (2002).

REVIEWERS' COMMENTS

Reviewer #1 (Remarks to the Author):

The authors added important analyses that better support the presence of disulfide bonds (Figure 2C, S3 and S6), the possible absence of C47-C182 disulfide bond upon contraction (Fig. 2E), and the molecular basis for the tail flexibility.

I would like to commend the authors for a fine study.

Note: Check for typos (e.g. l. 660; incomplete Ref37).

Reviewer #2 (Remarks to the Author):

The Authors addressed the lack of 'hard-evidence' demonstrating that phage Milano tail is crosslinked. The biological significance of this observation, the mechanisms of disulfide bond formation/oxidation and how this process relates to tail contraction are unknown.

Overall this paper reports an interesting phenomenological observation, similar to what recently reported for Pam3, but fails to give a biological explanation for how/why/when this phage would be crosslinked.

The take-home message is somewhat limited. It is also surprising that the authors diluted the already modest significance of this work by publishing a second standalone paper focused on phage Milano head and neck in Communications Biology. This paper also focuses on crosslinking "within and between neck, collar, capsid and tail that provide an exceptional structural stability to Milano" (from the abstract).

Reviewer #1 (Remarks to the Author):

The authors added important analyses that better support the presence of disulfide bonds (Figure 2C, S3 and S6), the possible absence of C47-C182 disulfide bond upon contraction (Fig. 2E), and the molecular basis for the tail flexibility.

I would like to commend the authors for a fine study.

Note: Check for typos (e.g. l. 660; incomplete Ref37).

Response:

Typos have been corrected.

Reviewer #2 (Remarks to the Author):

The Authors addressed the lack of 'hard-evidence' demonstrating that phage Milano tail is crosslinked. The biological significance of this observation, the mechanisms of disulfide bond formation/oxidation and how this process relates to tail contraction are unknown.

Overall this paper reports an interesting phenomenological observation, similar to what recently reported for Pam3, but fails to give a biological explanation for how/why/when this phage would be crosslinked.

The take-home message is somewhat limited. It is also surprising that the authors diluted the already modest significance of this work by publishing a second standalone paper focused on phage Milano head and neck in Communications Biology. This paper also focuses on crosslinking "within and between neck, collar, capsid and tail that provide an exceptional structural stability to Milano" (from the abstract).

Response: We are puzzled by these comments. For the initial submission of the paper, this reviewer wrote:

The paper by Sonani et al. describes an elegant structural analysis of the Agrobacterium tumefaciens bacteriophage Milano. The authors determined a cryo-EM reconstruction of the contractile tail of phage Milano in the bent and contracted conformations, deciphering the bent-to-straight transformation of the sheath and tube proteins. Unexpectedly, the authors found that the Milano tail sheath and baseplate are covalently linked by multiple disulfide bounds, which may enable the phage to survive the mechanical stress by binding to the host flagella. **Overall, these findings broaden our understanding of covalent crosslinking in bacteriophages and shed light on the biology of flagellotropic phages that do not carry a curly fiber to wrap around the host flagellum. Overall, I believe this paper has many strengths and advances the field of bacteriophage biology, especially for flagellotropic phages that are poorly understood. At the same time, there are several gaps that the authors should address to make this work publication quality. [our emphasis]**

The main gap involved the lack of additional evidence for the disulfides, such as by using mass spectrometry. We have done that, but the reviewer now says the paper "fails to give a biological explanation for how/why/when this phage would be crosslinked."

We thus think that this reviewer continues to raise questions that are way beyond the scope of the present paper.